# How explainable are adversarially-robust CNNs?

## Abstract

Three important criteria of existing convolutional neural networks (CNNs) are (1) test-set accuracy; (2) out-of-distribution accuracy; and (3) explainability *via feature attribution*. While these criteria have been studied independently, their relationship is unknown. For example, do CNNs with better out-of-distribution performance also have better explainability? Furthermore, most prior feature attribution studies only evaluate methods on 2-3 common vanilla ImageNet-trained CNNs, leaving it unknown how these methods generalize to CNNs of other architectures and training algorithms. Here, we perform the first large-scale evaluation of the relations of the three criteria using nine feature-attribution methods and 12 ImageNet-trained CNNs that are of three training algorithms and five CNN architectures. We find several important insights and recommendations for ML practitioners. First, adversarially robust CNNs have a higher explainability score on gradient-based attribution methods (but not CAM-based or perturbation-based methods). Second, AdvProp models, despite being highly accurate, are not superior in explainability. Third, among nine feature attribution methods tested, GradCAM and RISE are consistently the best methods. Fourth, Insertion and Deletion are biased towards vanilla and robust models respectively, due to their strong correlation with the confidence score distributions of a CNN. Fifth, we did not find a single CNN to be the best in all three criteria, which suggests that CNNs with better performance do not have better explainability. Sixth, ResNet-50 is on average the best architecture among the architectures used in this study, which indicates architectures with higher test-set accuracy do not necessarily have better explainability scores.

## 1 Introduction

A key property of trustworthy and explainable of image classifiers is their capability of leveraging non-spurious features, i.e., "right for the right reasons" Alcorn et al. (2019); Lapuschkin et al. (2016). To assess this property of a convolutional neural network (CNN), attribution maps (AMs) (Bansal et al., 2020; Agarwal & Nguyen, 2020) are commonly used as they show the input pixels that are important for or against a predicted label (Fig. 3). AMs, a.k.a. "saliency maps", have been useful in many tasks including localizing malignant tumors in x-ray images (Rajpurkar et al., 2017), detecting objects in a scene (Zhou et al., 2016), discovering biases in image classifiers (Lapuschkin et al., 2016), and explaining image similarity (Hai Phan, 2022). Most AM methods are post-hoc explanation methods and can be conveniently applied to any CNNs. Despite their wide applicability, state-of-the-art AM methods, e.g. (Zhou et al., 2016; Agarwal & Nguyen, 2020; Selvaraju et al., 2017), are often evaluated on the same set of ImageNet-trained vanilla classifiers (mostly AlexNet (Krizhevsky et al., 2012), ResNet-50 (He et al., 2016), or VGG networks (Simonyan & Zisserman, 2014)). Such limited evaluation raises an important question: **How do AM methods generalize to other models that are of different architectures, of different accuracy, or trained differently?**

In this paper, we analyze the practical trade-offs among three essential properties of CNNs: (1) test-set accuracy; (2) out-of-distribution (OOD) accuracy (Alcorn et al., 2019; Nguyen et al., 2015); and (3) explainability via AMs a.k.a. feature attribution maps (Bansal et al., 2020). We perform the *first* large-scale, empirical, multi-dimensional evaluation of nine state-of-the-art AM methods, each on 12 different models for five different CNN architectures. Using four common evaluation metrics, we assess each AM method on five vanilla classifiers and five adversarially robust models, i.e. of the same architectures but trained with

an adversarial training method (Xie et al., 2020; Madry et al., 2017)—a leading training framework that substantially improves model accuracy on OOD data. Comparing vanilla and robust models is important because robust models tend to generalize better on some OOD benchmarks (Chen et al., 2020; Zhang & Zhu, 2019; Xie et al., 2020; Yin et al., 2019) but at the cost of worse test-set accuracy (Madry et al., 2017; Zhang et al., 2019). Robust models admit much smoother (i.e. more interpretable) gradient images (Bansal et al., 2020; Zhang & Zhu, 2019; Etmann et al., 2019), which questions whether robust models are more *explainable*[1] than their vanilla counterparts when using state-of-the-art AM methods. Our key findings are:

1. Robust models are more explainable than vanilla ones under gradient-based AMs but not CAM-based or perturbation-based AMs (Sec. 3.1).

2. On all 12 models and under four different AM evaluation metrics, GradCAM (Selvaraju et al., 2017) and RISE (Petsiuk et al., 2018) are consistently among the three best methods (Sec. 3.2). Thus, considering both time complexity and explainability, CAM and GradCAM are our top recommendations, which suggest that the field's progress since 2017 may be stagnant.

3. None of the CNNs are best in all three criteria: test-set accuracy, accuracy on adversarial examples, and explainability. For example, the models with the highest test-set accuracy, e.g. DenseNet, do not have the highest localization scores. We also find a competitive ResNet-50 CNN trained via AdvProp (PGD-1) to be an all-around winner under three explainability metrics: Pointing Game, WSL, and Insertion (Sec. 3.4).

4. On average, ResNet-50 scores higher than all other five architectures used in this study across all metrics. This suggests that architectures with higher test-set accuracy do not necessarily have better explainability scores considering that DenseNet-161 has the highest test-set accuracy. (Sec. 3.3)

5. In contrast to vanilla and robust models, which are trained on either real and adversarial images, respectively (Madry et al., 2017; Zhang et al., 2019), CNNs trained on *both* real and adversarial data via AdvProp (Xie et al., 2020) are highly competitive on both test-set and OOD data. Yet, we find AdvProp models to be *not superior* in explainability compared to robust models (Sec. 3.5).

6. Insertion (Samek et al., 2016) is heavily biased towards vanilla models while Deletion strongly favors robust models, which tend to output more conservative confidence scores. In general, these two score-based metrics strongly correlate with the mean confidence scores of classifiers and largely disagree (Secs. 3.6 & 3.7).

## 2 Methods and experimental setup

### 2.1 Datasets

Consistent with prior work (Fong & Vedaldi, 2017; Fong et al., 2019; Zhou et al., 2016), we use the 50K-image ILSVRC 2012 validation set (Russakovsky et al., 2015) to evaluate AM methods. First, to filter out ambiguous multi-object ImageNet images (Beyer et al., 2020), we remove images that have more than one bounding box. Furthermore, to more accurately appreciate the quality of the heatmaps, we remove the parts whose bounding box covers more than 50% of the image area because such large objects do not allow to distinguish AM methods well (on ImageNet, a simple Gaussian baseline already obtains ∼52% weakly-supervised localization accuracy (Choe et al., 2020)).

For the remaining images, as Bansal et al. (2020), we evaluate AMs on two ImageNet subsets: (a) 2,000 random validation-set images; (b) 2,000 images that a pair of (vanilla, robust) models correctly label for each architecture (hereafter, ImageNet-CL) to understand the explainability of models in general vs. when they make correct predictions.

### 2.2 Image classifiers

We run AM methods on five ImageNet-pretrained CNN classifiers of five different architectures: AlexNet (Krizhevsky et al., 2012), GoogLeNet (Szegedy et al., 2015), ResNet-50 (He et al., 2016), DenseNet-161 (Huang et al., 2017), and MobileNet-v2 (Sandler et al., 2018). To understand the relationship between

---

[1]"Explainable" here means scoring high on the four AM evaluation metrics in Sec. 2.4.

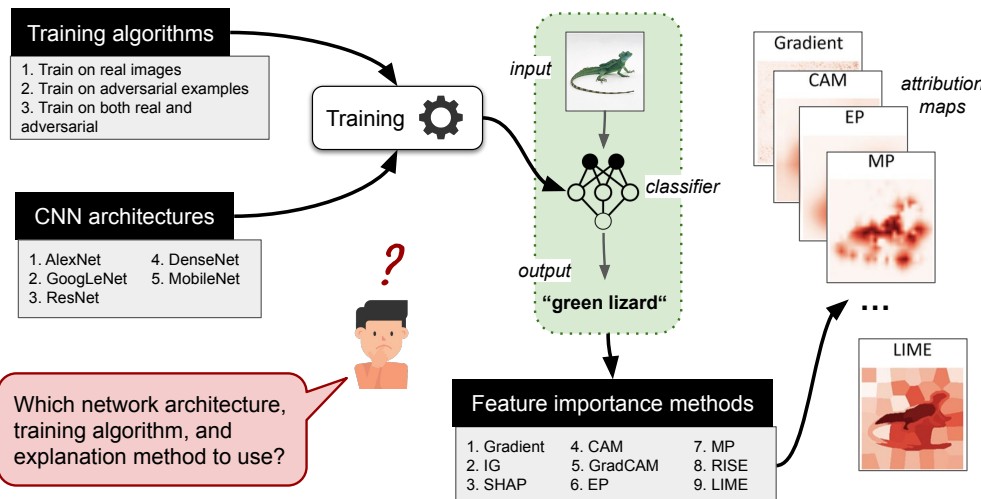

Figure 1: Given a pair of training algorithm and a CNN architecture, the training process often results in a *distinct* classifier of unique test-set classification accuracy, out-of-distribution accuracy (Table 1), and network properties (Chen et al., 2020). Furthermore, each classifier's explainability via attribution maps may vary greatly depending on the network itself and also the feature attribution method of choice (Fig. 3). Our work is the first to explore this complex but practical model-selection space in machine learning and provide a set of recommendations to practitioners.

adversarial accuracy and explainability, we compare the vanilla models with those trained via adversarial training (Madry et al., 2017). For completeness, we also evaluate AM methods on ResNet-50 models trained on a mix of real and adversarial data via AdvProp (Xie et al., 2020). The robust models tend to have ∼1.8× lower confidence scores, lower ImageNet accuracy, but much higher adversarial accuracy[2] than vanilla models (Table 1). In contrast, AdvProp-trained models have similar ImageNet accuracy and confidence scores to vanilla models, but higher adversarial accuracy than robust models (Table 1; PGD-1 & PGD-5).

## 2.3 Feature attribution methods

**Gradient-based methods** **Gradient** (Simonyan et al., 2013) uses the gradient image, which measures the sensitivity of the confidence score of a target label to changes in each pixel. **Integrated Gradients** (IG) (Sundararajan et al., 2017) ameliorates the gradient saturation problem by linearly interpolating between the input image and a reference zero image and averaging the gradients over all the interpolation samples. **SHapley Additive exPlanations** (SHAP) (Lundberg & Lee, 2017a) approximates Shapley values of a prediction by assessing the effect of deleting a pixel under all possible combinations of the presence and absence of the other pixels. While SHAP is theoretically grounded on the classic Shapley value, it often requires a larger sample size than IG (200 vs. 50) and is thus slower.

**CAM-based methods** **CAM** (Zhou et al., 2016) produces a heatmap by taking a weighted average of the channels at the last convolutional layer of a CNN where the channel weights are the weights of the fully-connected layer that connects the global averaged pooling (GAP) value of each channel to the network classification outputs. CAM is only applicable to CNN architectures with the GAP layer right before the last classification layer. **GradCAM** (Selvaraju et al., 2017) approximates a channel's weight in CAM by the mean gradient over the activations in that channel and is applicable to all CNNs.

**Perturbation-based methods** **Meaningful Perturbation** (MP) (Fong & Vedaldi, 2017) finds the smallest real-valued, Gaussian-blurred mask such that when applied to the input image it minimizes the target confidence score. MP is sensitive to changes in hyperparameter values (Bansal et al., 2020). To ameliorate the sensitivity to hyperparameters, Fong et al. (2019) propose to average over four MP-like

---

[2]Following Chen et al. (2020); Bansal et al. (2020), our PGD attacks use seven steps per image; $L_2$ norm $\epsilon = 3$, and a step size $= 0.5$.

heatmaps of varying controlled sizes of the high-attribution region. **Extremal Perturbation** (EP) (Fong et al., 2019) is less sensitive to hyperparameter values than MP but requires 4 separate optimization runs to generate one AM.

While MP and EP are only applicable to models that are differentiable, **LIME** (Ribeiro et al., 2016) and **RISE** (Petsiuk et al., 2018) are two popular AM methods that are applicable to black-box CNNs, where we only observe the predictions. LIME generates $N$ masked images by zeroing out a random set of non-overlapping superpixels and takes the mean score over the masked samples where the target superpixel is *not* masked out as the attribution. Like CAM, RISE also takes a weighted average of all channels but the weighting coefficients are the scores of randomly-masked versions of the input image.

Table 1: Top-1 accuracy (%) of pre-trained networks and their confidence scores on ImageNet-CL.

| Model name | Architecture | Training | Confidence | ImageNet acc. | Adv acc. |
|---|---|---|---|---|---|
| AlexNet | AlexNet (Krizhevsky et al., 2012) | vanilla | $0.79 \pm 0.21$ | 56.55 | 0.18 |
| AlexNet-R | AlexNet (Krizhevsky et al., 2012) | PGD (Madry et al., 2017) | $0.41 \pm 0.29$ | 39.83 | 22.27 |
| GoogleNet | GoogLeNet (Szegedy et al., 2015) | vanilla | $0.78 \pm 0.22$ | 69.78 | 0.08 |
| GoogleNet-R | GoogLeNet (Szegedy et al., 2015) | PGD (Madry et al., 2017) | $0.49 \pm 0.29$ | 50.94 | 31.23 |
| ResNet | ResNet-50 (He et al., 2016) | vanilla | $0.89 \pm 0.16$ | 76.15 | 0.35 |
| ResNet-R | ResNet-50 (He et al., 2016) | PGD (Madry et al., 2017) | $0.56 \pm 0.30$ | 56.25 | 36.11 |
| PGD-5 | ResNet-50 (He et al., 2016) | AdvProp (Xie et al., 2020) | $0.90 \pm 0.15$ | 77.01 | 73.55 |
| PGD-1 | ResNet-50 (He et al., 2016) | AdvProp (Xie et al., 2020) | $0.91 \pm 0.14$ | 77.31 | 69.02 |
| DenseNet | DenseNet-161 (Huang et al., 2017) | vanilla | $0.90 \pm 0.18$ | 77.14 | 0.51 |
| DenseNet-R | DenseNet-161 (Huang et al., 2017) | PGD (Madry et al., 2017) | $0.66 \pm 0.34$ | 66.12 | 41.76 |
| MobileNet | MobileNet-v2 (Sandler et al., 2018) | vanilla | $0.87 \pm 0.23$ | 71.88 | 0.01 |
| MobileNet-R | MobileNet-v2 (Sandler et al., 2018) | PGD (Madry et al., 2017) | $0.41 \pm 0.26$ | 50.40 | 29.65 |

## 2.4 Explanability under four feature-attribution evaluation metrics

**Pointing Game** (PG) considers an AM correct if the highest-attribution pixel lies inside the human-annotated bounding box; i.e. the main object in an image must be used by CNNs to predict the ground-truth label. We use the PG implementation of TorchRay (Fong & Vedaldi, 2019).

**Weakly Supervised Localization** (WSL) (Zhou et al., 2016; Bau et al., 2017) is a more fine-grained version of PG. WSL computes the Intersection over Union (IoU) of the human-annotated bounding box and bounding box generated from an AM via threshold-based discretization. We use the WSL evaluation framework by (Choe et al., 2020) and consider an AM correct if the IoU > 0.5.

*Assumptions:* Both PG and WSL are standard metrics in evaluating AMs (Zhou et al., 2016; Bau et al., 2017; Adebayo et al., 2018; Fong & Vedaldi, 2017). While PG is a more coarse metric and WSL a more fine-grained version, both are based on the assumption that the discriminative signals for CNNs to label an image are on the main object annotated by humans.

The **Deletion** metric (Petsiuk et al., 2018; Samek et al., 2016; Gomez et al., 2022) ($\downarrow$ lower is better) measures the Area under the Curve (AUC) of the target-class probability as we zero out the top-$N$ highest-attribution pixels at each step in the input image. That is, a faithful AM is expected to have a lower AUC in Deletion. For the **Insertion** metric (Samek et al., 2016; Gomez et al., 2022; Petsiuk et al., 2018) ($\uparrow$ higher is better) we start from a zero image and add top-$N$ highest-attribution pixels at each step until recovering the original image and calculate the AUC of the probability curve. For both Deletion and Insertion, we use the implementation by (Petsiuk, 2018) and $N = 448$ at each step.

*Assumptions:* The Deletion and Insertion metrics (Petsiuk et al., 2018) are based on two assumptions. First, it assumes that each pixel is an independent variable. Therefore, it is possible to change a model's predictions by removing or adding one pixel. Second, it assumes that removing a pixel by replacing it with a zero (i.e. gray) pixel is a reasonable removal operator that yields a counterfactual sample near or on the true data manifold (Covert et al., 2020).

**Hyperparameter optimization**     Perturbation-based methods have explicit numeric hyperparameters and their performance can be highly sensitive to the values (Bansal et al., 2020). Using grid search, we tune the hyperparameters for each pair (AM method, CNN). See Sec. A.1 for tuned hyperparameters and optimized values.

## 3    Experimental results

Given a pair of training algorithm and a CNN architecture, the training process often results in a *distinct* classifier of unique test-set classification accuracy, out-of-distribution accuracy (Table 1), and network properties Chen et al. 2020. Furthermore, each classifier's explainability via attribution maps may vary greatly depending on the network itself and also the feature attribution method of choice (Fig. 3). For the first time in the literature, we shed light on this complex model-selection space by addressing the following questions:

1. Do adversarially-robust models provide better attribution maps? (Sec. 3.1)
2. Which are the best feature importance methods (over all architectures and training algorithms)? That is, which feature importance method should an ML engineer use to visualize an arbitrary image classifier? (Sec. 3.2)
3. Which network *architecture* yields the best feature-attribution maps? (Sec. 3.3)
4. Which is the all-around best model given classification accuracy, explainability via feature attribution, and classification runtime? (Sec. 3.4)
5. Does training on both real and adversarial images improve model accuracy *and* feature-attribution-based explainability? (Sec. 3.5)
6. Finally, our novel comparison between vanilla and adversarially-robust models reveals important findings that the common Insertion and Deletion metrics *strongly* correlate with classification confidence scores (Sec. 3.6) and *weakly* correlate with the other common evaluation paradigm of WSL (Sec. 3.7).

### 3.1    Robust models provide better gradient-based AM explanations than vanilla models

Robust models often achieve better accuracy than vanilla CNNs on OOD images (Chen et al., 2020; Xie et al., 2020; Madry et al., 2017; Agarwal et al., 2019). In terms of interpretability, robust models admit smoother gradient images (Bansal et al., 2020) and smoother activation maps, i.e. they internally filter out high-frequency input noise (Chen et al., 2020). As these two important properties affect the quality of AMs, we test whether AMs generated for robust models are better than those for vanilla models.

**Experiments**     For each of the 12 CNNs, we run the nine AM methods on 2,000 ImageNet-CL images. That is, for each CNN, we generate $9 \times 2,000 = 18,000$ images (for MobileNet and AlexNet CNNs, we only run eight methods as CAM is not applicable to architectures that have no Global Average Pooling(GAP)

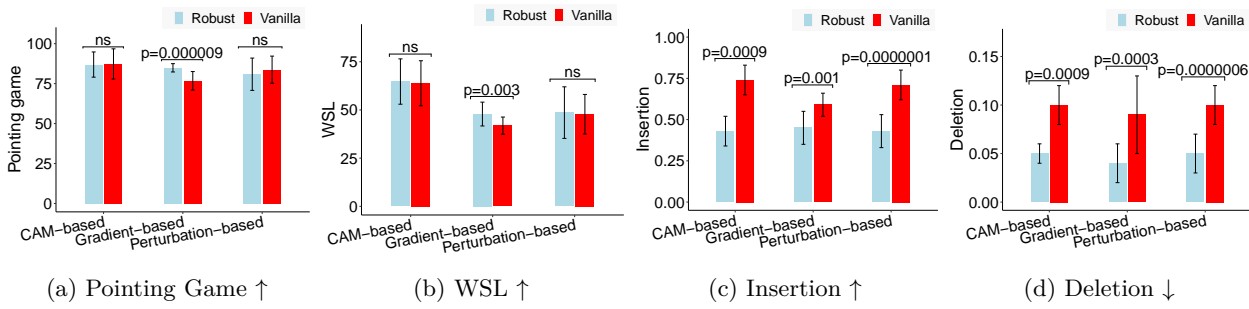

| (a) Pointing Game ↑ | (b) WSL ↑ | (c) Insertion ↑ | (d) Deletion ↓ |

Figure 2: The AMs of robust models consistently score higher than those of vanilla models on Pointing Game (Zhang et al., 2018) (a) and weakly-supervised localization (WSL) (Zhou et al., 2016) (b). For Deletion (↓ lower is better), robust models consistently outperform vanilla models for all three groups of AM methods (d). In contrast, under Insertion, vanilla models always score better (c).

layer). For each AM, we compute the four evaluation metrics and compare the distribution of scores between vanilla and robust models (detailed scores per CNN are in Sec. A.5).

**Localization results** For PG and WSL, we find *no significant differences* between vanilla and robust AMs (Fig. 2; a–b) except for the gradient-based AMs where robust CNNs significantly outperform vanilla models (Mann Whitney U-test; $p$-value < 0.003). CAM-based AMs of these two types of CNNs are visually similar for most images (see Fig. 3a and d). The perturbation-based AMs are less similar and neither type of model outperforms the other significantly (Fig. 3). For all three gradient-based methods, the robust models' AMs often clearly highlight the main object (Fig. 3), which is consistent with SmoothGrad AMs (Zhang & Zhu, 2019).

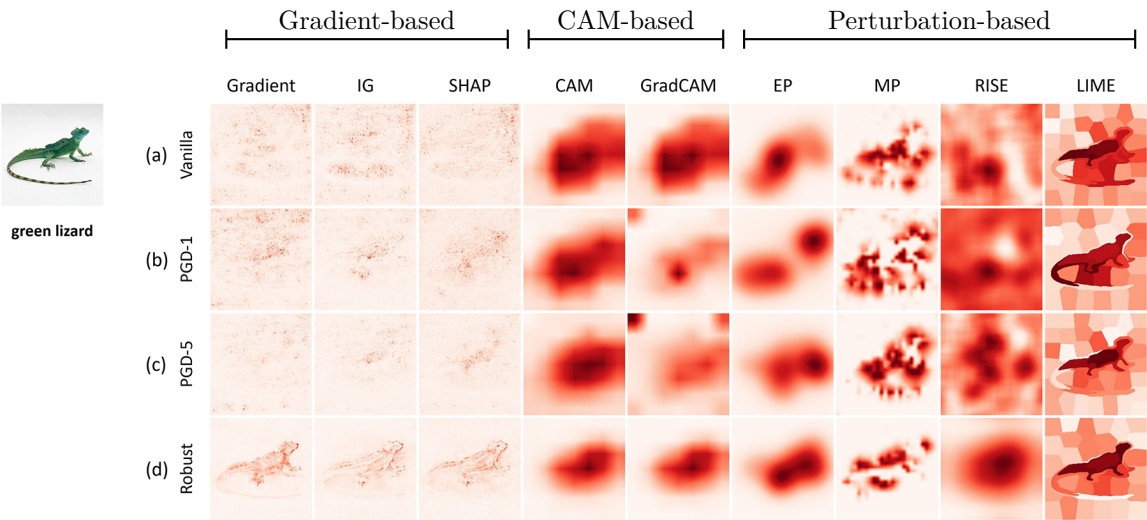

Figure 3: Comparison of attribution maps generated by nine AM methods for four different ResNet-50 models for the same input image and target label of *green lizard*. From top down: (a) vanilla ImageNet-trained ResNet-50 (He et al., 2016); (b–c) the same architecture but trained using AdvProp (Xie et al., 2020) where adversarial images are generated using PGD-1 and PGD-5 (i.e. 1 or 5 PGD attack steps (Madry et al., 2017) for generating each adversarial image); and (d) a robust model trained exclusively on adversarial data via the PGD framework (Madry et al., 2017). The more CNNs are trained on adversarial perturbations (top down), the less noisy and more interpretable AMs by gradient-based methods tend to be. See Sec. A.7 for more examples.

**Score-based results** For all methods, we find a significant difference between the AMs of robust and those of vanilla models (Mann Whitney U-test; $p$-value < 0.002). For Insertion, vanilla models outperform robust models, but robust models outperform vanilla models for Deletion (Fig. 2).

The results for ImageNet are similar to ImageNet-CL (Fig. 12). Robust models have higher accuracy on adversarial examples and higher explainability under gradient-based AMs, but not CAM-based or perturbation-based AMs.

**Recommendation** ML practitioners should use CAM-based methods for either vanilla or robust models as there is no difference in explainability. However, CAM-based methods have fewer hyperparameters and run faster than perturbation-based AM methods.

## 3.2 GradCAM and RISE are the best feature attribution methods

Usually, AM methods are only compared to a few others and only on vanilla ImageNet-trained CNNs (Fong et al., 2019; Petsiuk et al., 2018; Zhou et al., 2016). It is therefore often unknown whether the same conclusions hold true for non-standard CNNs that are trained differently or have different architectures.

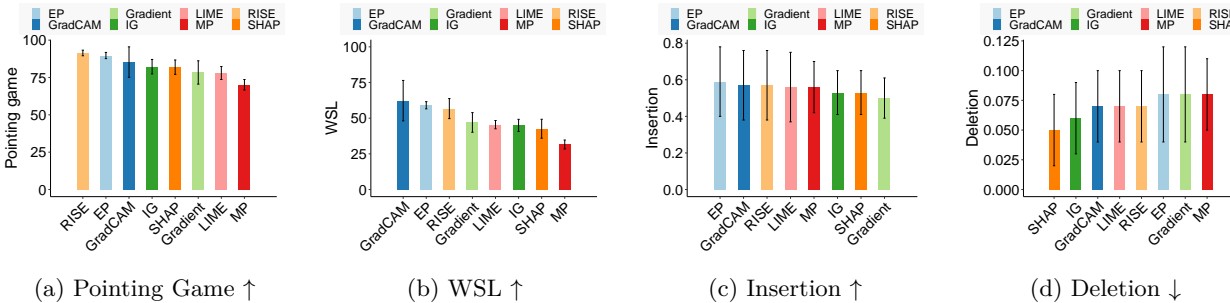

|  |  |  |  |
|---|---|---|---|
| (a) Pointing Game ↑ | (b) WSL ↑ | (c) Insertion ↑ | (d) Deletion ↓ |

Figure 4: The average performance of 8 attribution methods across 10 CNNs show that GradCAM and RISE are among the top-3 for Pointing Game (a), WSL (b), Insertion (c), and Deletion (d) (↓ lower is better) while MP is the worst on average across all four metrics (see Table 4). The results for ImageNet are shown in Fig. 15.

**Experiments** For each of the 8 AM methods (excluding CAM, which is not applicable to AlexNet and MobileNet), we average across the ten CNNs (Sec. 2.2), i.e. 5 vanilla and 5 robust CNNs, excluding two AdvProp models.

**Results** Fig. 4 shows that in terms of average ranking, shown in Table 4, GradCAM and RISE are the best methods overall, closely followed by EP. Since the quantitative and qualitative results of CAM and GradCAM are almost identical (Sec. A.5 and Fig. 3), CAM would also be among the best AM methods, if applicable to a given CNN. MP is the worst AM method on average. Besides the explainability, Table 2 shows the time complexity and runtime of the methods; GradCAM is the overall best.

In the literature, the simple Gradient (Simonyan et al., 2013) method often yields noisy saliency maps (Smilkov et al., 2017; Sundarararajan et al., 2017) on vanilla CNNs and is thus regarded as one of the worst AM methods. When averaging its performance over all 10 CNNs (both vanilla and robust), the simple Gradient method outperforms the more complicated methods LIME, IG, SHAP, and MP in terms of both WSL (Fig. 4b) and runtime. Again the results for ImageNet are similar to those on ImageNet-CL (Fig. 15, Tables 4 & 5).

**Recommendation** Given an arbitrary CNN, we recommend using GradCAM.

### 3.3 ResNet-50's explanations outperform other architectures on average

We used the five different architectures described in Sec. 2.2 for our experiments. We investigate which architecture has the best performance overall for all evaluation metrics. To the best of our knowledge, we are the first to consider explainability when comparing architectures.

**Experiment** Given the experiment in Sec. 3.1, we consider each architecture (including vanilla and robust CNNs) across our nine attribution methods on the ImageNet-CL data set. For each metric, we show the boxplot of each architecture (excluding AdvProp networks to have a fair comparison of architectures as other architectures only consist of robust and vanilla). Fig. 5 depicts the average ranking of the explanations derived from different architectures over robust and vanilla networks and eight attribution methods (excluding CAM) across our four metrics.

**Results** Fig. 5 shows that ResNet-50 is the best architecture in terms of average ranking. GoogLeNet and DenseNet-161 are two competitive architectures after ResNet-50 while MobileNet-v2 and AlexNet are the worst architectures overall. Even though ResNet-50 is the best on average, it is one of the worst in terms of Deletion. In the same vein, AlexNet is the best in Deletion but is the worst in both localization metrics, i.e. WSL and pointing game.

The ImageNet results shown in Fig. 14 allow to draw the same conclusion as ImageNet-CL regarding architectures, i.e. ResNet-50 and AlexNet are on average the best and the worst architectures, respectively.

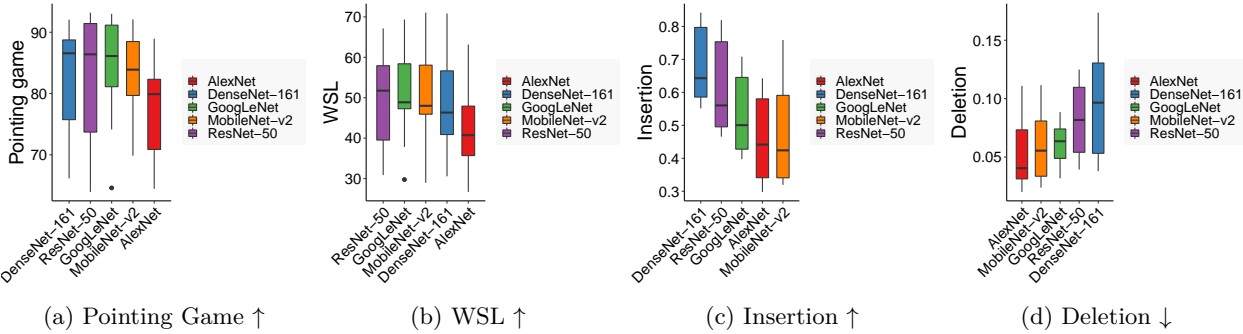

(a) Pointing Game ↑     (b) WSL ↑     (c) Insertion ↑     (d) Deletion ↓

Figure 5: (a)-(d) show the average performance of our five architectures in Sec. 2.2 across the eight attribution methods in Sec. 2.3 (excluding CAM) for pointing game, WSL, Insertion, and Deletion, respectively. On average, RestNet-50 is the best overall, and AlexNet is the worst (see Table 7).

Our findings suggest that architectures with higher test-set accuracy do not necessarily have better explainability scores; even though DenseNet-161 has the highest average accuracy, it is not the best architecture across our evaluation metrics on average (Table 7).

**Recommendation** We recommend ML practitioners use the ResNet-50 architecture as it gives the best average performance for top-1 classification accuracy and explainability.

### 3.4 Which is the all-around best model given the Pareto front of classification accuracy, explainability via feature attribution, and classification runtime?

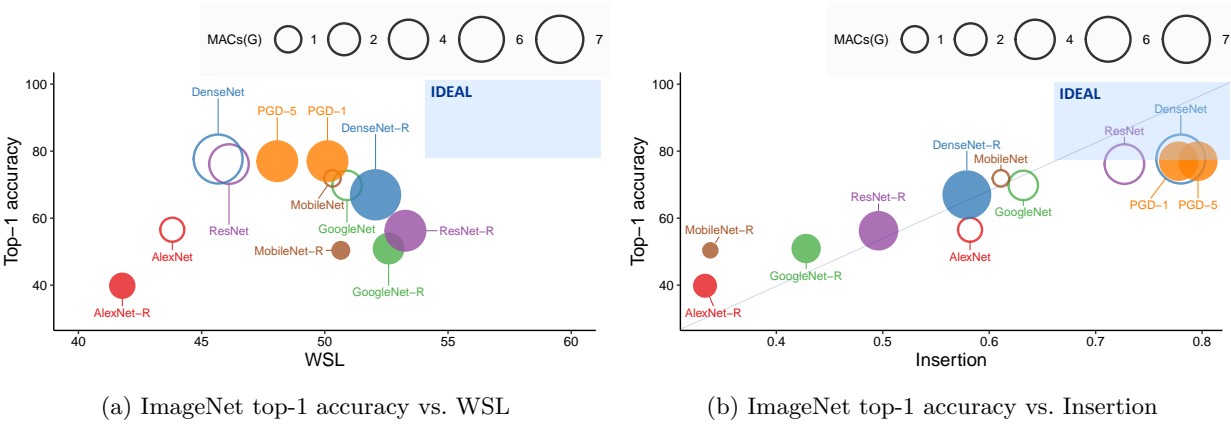

(a) ImageNet top-1 accuracy vs. WSL     (b) ImageNet top-1 accuracy vs. Insertion

Figure 6: The average performance of all 12 CNNs across eight attribution methods under WSL and Insertion, showing ImageNet top-1 accuracy and MACs. Filled circles denote robust and empty circles vanilla models. No network is consistently the best across all metrics and criteria. Fig. 9 shows the results for the Pointing Game and Deletion metrics. The IDEAL region is where ideal CNNs of the highest classification and explanation performance are.

It is a practical question but under-explored in the literature: Which is the all-around best CNN w.r.t. classification accuracy, explainability, and runtime. Our comprehensive experimental evaluation allows us to investigate this question for the first time.

**Experiments** From the experimental results described above, we compare the ImageNet validation classification accuracy (see Table 1), explainability scores, and the number of multiply–accumulate operations (MACs) (He et al., 2016) (measured with the method by Sovrasov (2019)) of all 12 CNNs.

**Results** In terms of WSL and Pointing Game, no CNN performs best in both classification accuracy and localization (Fig. 6a; the IDEAL region is empty). Higher accuracy is not correlated with better localization and DenseNet-R, PGD-1, and ResNet-R are on the Pareto front. Insertion strongly correlates with the classification accuracy of CNNs (Fig. 6b) and the best CNNs under Insertion are DenseNet, PGD-1, and PGD-5. While no CNN is the best for all four metrics, we find PGD-1 to perform the best among 12 CNNs under Pointing Game, WSL, and Insertion (see Fig. 16).

**Recommendation** ML practitioners should use AdvProp-trained models to maximize the classification accuracy and explainability, here PGD-1 for the ResNet-50 architecture.

### 3.5 Training on both real and adversarial data improves predictive performance and explainability

Many high-stake applications, e.g. in healthcare, require CNNs to be both accurate and explainable (Lipton, 2017). AdvProp models, which are trained on both real and adversarial data, obtain high accuracy on both test-set and OOD data (Xie et al., 2020)—better than both vanilla and robust models. We test whether AdvProp CNNs offer higher explainability than vanilla and robust models.

**Experiments** We follow our setup above (Sec. 3.1) for four CNNs of the same ResNet-50 architecture: vanilla ResNet, PGD-1, PGD-5, and ResNet-R, and compute all four metrics for ImageNet-CL.

**Results** The robust models have the best accuracy for object localization (Fig. 7; a–b); vanilla models score the worst. AdvProp models are consistently performing in between vanilla and robust models; better than vanilla models under Insertion (Fig. 7c), and similar to vanilla models under Deletion (Fig. 7d). The results for ImageNet are similar to those of ImageNet-CL (see Fig. 13).

**Recommendation** We recommend to always use AdvProp models instead of vanilla models as AdvProp models achieve not only higher classification accuracy than vanilla models but also offer better explainability under Pointing Game, WSL, and Insertion (Fig. 7a–c).

Table 2: AM methods ranked by average WSL performance over 10 CNNs (excluding AdvProp models), runtime complexity, number of forward passes, number of backward passes, default hyperparameters, and median runtime per image.

| Methods | WSL | # of forward passes | # of backward passes | $O(n)$ | Hyperparameters | Runtime (ms) |
|---|---|---|---|---|---|---|
| **CAM** | $67.63 \pm 1.67$ | 1 | 0 | $O(1)$ | n/a | 108 |
| **GradCAM** | $62.29 \pm 14.17$ | 1 | 1 | $O(1)$ | n/a | 34 |
| **EP** | $59.06 \pm 2.53$ | i (# iterations) × a (# areas) | i (# iterations) × a(# areas) | $O(n)$ | N = 800, a = 4 | 32,063 |
| **RISE** | $56.72 \pm 7.03$ | N (# samples) | 0 | $O(n)$ | N = 8,000 | 8,485 |
| **Gradient** | $47.03 \pm 6.88$ | 1 | 1 | $O(1)$ | n/a | 26 |
| **LIME** | $45.47 \pm 2.91$ | N (# samples) | 0 | $O(n)$ | N = 1,000 | 7,234 |
| **IG** | $44.97 \pm 4.23$ | N (# samples) | N (# samples) | $O(n)$ | N = 50 | 196 |
| **SHAP** | $42.61 \pm 6.63$ | N (# samples) | N (# samples) | $O(n)$ | N = 200 | 1,137 |
| **MP** | $31.58 \pm 3.12$ | i (# iterations) | i (# iterations) | $O(n)$ | N = 500 | 7,530 |

### 3.6 Insertion and Deletion scores are predictable from the confidence scores, regardless of the feature-attribution explanation quality

Insertion and Deletion are among the most common AM evaluation metrics in both computer vision (Petsiuk et al., 2018; Agarwal & Nguyen, 2020; Erion et al., 2021; Hooker et al., 2019; Samek et al., 2016; Tomsett et al., 2020) and natural language processing (Pham et al., 2021; Kim et al., 2020; Arras et al., 2017; DeYoung et al., 2020). Yet, our results comparing vanilla and robust models show the surprising phenomenon that all robust CNNs consistently outperform vanilla CNNs under Deletion, but the reverse is true under Insertion (Figs. 2 and 16c). We investigate whether this is because robust models tend to have much lower confidence than vanilla models (Table 1). That is, Insertion and Deletion scores might be simply reflecting the confidence score distribution of a CNN rather than the true explainability of the AMs.

**Experiments** We calculate the Spearman-rank correlation between the confidence score and Deletion and Insertion score at both the architecture and image levels. At the architecture level, we compute the Spearman-rank correlation coefficient between each CNN's average confidence (over 2,000 ImageNet images)

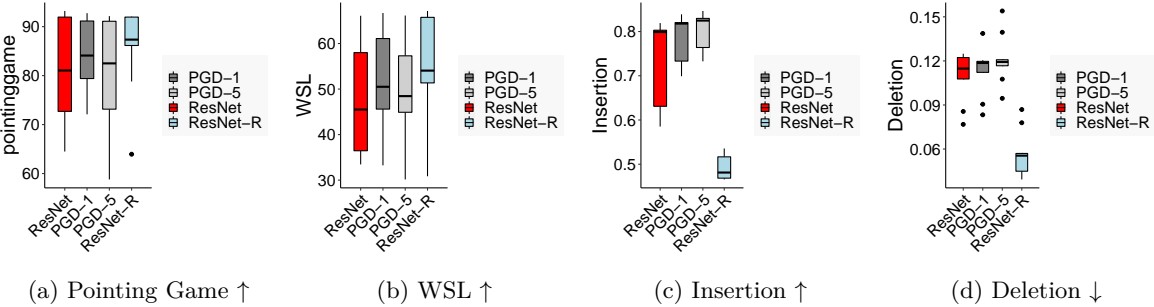

Figure 7: Averaging across 9 AM methods, AdvProp models (PGD-1 and PGD-5) outperform vanilla models (ResNet) but are worse than robust models (ResNet-R) in two object localization metrics (a & b). See Fig. 3 for qualitative results that support AdvProp localization ability. For Insertion, AdvProp models are also better than vanilla. The same holds for ImageNet images (Fig. 13).

and their average Deletion (or Insertion) scores. At the image level, we compute the same correlation between a single confidence score (given a CNN) and a Deletion score for each image for all 10 CNNs and 9 AM methods.

**Results**   We find a strong correlation of 0.88 between CNN mean confidence score and mean Deletion score (Fig. 8a VI), and also a strong Spearman-rank correlation of 0.78 between an image's confidence score and Deletion score (for all 10 CNNs, all 2,000 images and all AMs). The same is true at the image level (Fig. 8a VI). See Fig. 8a for qualitative results illustrating that the Deletion scores are mostly predictable from the confidence scores alone, regardless of the actual heatmap quality.

**Recommendation**   We do not recommend Insertion and Deletion for assessing the explainability of models that have dissimilar confidence-score distributions.

### 3.7   Insertion and Deletion can largely disagree and weakly correlate with localization metrics

Evaluating AMs is an open question (Doshi-Velez et al., 2017; Doshi-Velez & Kim, 2017) and no automatic evaluation metric is yet a single best estimator of AM faithfulness. Therefore, the community uses suites of metrics (four of them in this paper). We quantify how these four metrics correlate with one another.

**Experiments**   We compute the Spearman-rank correlation coefficient for each pair of metrics from all the AMs generated for ImageNet-CL images using all 12 CNNs and 9 methods.

**Results**   When CNNs correctly classify images, Pointing Game and WSL strongly and positively correlate (Spearman-rank correlation of 0.82). These two localization metrics only weakly correlate with score-based metrics (Table. 8b; from -0.09 to 0.18). Insertion and Deletion are strongly correlated as they are both strongly correlated with confidence scores (Table. 8b; 0.86). A heatmap considered good under Insertion is highly likely to be considered bad under Deletion, and vice versa. We hypothesize that this divergence between Insertion and Deletion is due to the fact that our set of networks contain both vanilla and robust models and that the two score-based metrics are reflecting the confidence-score distributions instead of the AM quality.

## 4   Related work

**Generalization of attribution methods**   Most AM studies in the literature (e.g. Zhou et al. (2016); Fong et al. (2019); Selvaraju et al. (2017); Chattopadhay et al. (2018)) only evaluate AM methods on two or three of the following vanilla ImageNet-pretrained CNNs: AlexNet, ResNet-50, GoogLeNet, and VGG-16. Recent work found that AMs on robust models are less noisy and often highlight the outline of the main object. Yet, such comparisons (Chen et al. (2020); Bansal et al. (2020); Zhang & Zhu (2019)) were done using only one AM method (vanilla Gradient in Bansal et al. (2020) or SmoothGrad in Zhang & Zhu (2019)). In

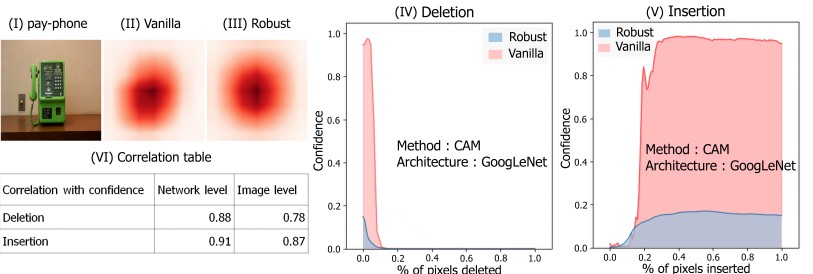

(a) Deletion (IV) and Insertion (V) scores disagree on which attribution map (II vs. III) is better.

|  | PG | WSL | Insertion | Deletion |
|---|---|---|---|---|
| **PG** | 1 | **0.82** | 0.18 | -0.10 |
| **WSL** | - | 1 | 0.08 | -0.09 |
| **Insertion** | - | - | 1 | **0.86** |
| **Deletion** | - | - | - | 1 |

(b) Spearman-rank correlation coefficient between evaluation metrics for ImageNet-CL. The correlation coefficients for ImageNet are slightly lower but exhibit the same trends (Fig. 16).

Figure 8: Insertion and Deletion scores are strongly correlated with a CNN's confidence scores for an image regardless of heatmap quality (VI). For the same pay phone image (I), the CAM heatmaps of GoogleNet and GoogleNet-R are visually similar (II and III). Yet, under Deletion, the robust heatmap (III) is better. In contrast, under Insertion, the vanilla heatmap (II) is better. The vanilla AUC is always larger than the robust AUC (IV and V). That is, Insertion and Deletion are not exclusively measuring the quality of heatmaps, yielding misleading conclusions. More examples in Sec. A.2.2.

contrast, our work is the first large-scale, systematic evaluation of nine AM methods on an unprecedentedly-large set of 12 different ImageNet CNNs spanning five CNN architectures and three training algorithms (see Table 1). Over 12 CNNs, we find GradCAM to be the best-performing (among nine tested AM methods) when considering all four metrics. This finding suggests that the dozens of newer AM methods in the past six years are not better than GradCAM in general, except for specific combinations of CNNs and evaluation metrics.

**Explainability of robust models** Robust models have more interpretable *gradient* images (Madry et al., 2017; Zhang & Zhu, 2019; Noack et al., 2021; Ross & Doshi-Velez, 2018) than those of vanilla CNNs. It is not known whether this is true for state-of-the-art AM methods.

**Score-based attribution evaluation metrics** For assessing AMs, Deletion, Insertion, and other confidence-score-based metrics are widely used in both computer vision (Petsiuk et al., 2018; Agarwal & Nguyen, 2020; Erion et al., 2021; Hooker et al., 2019; Samek et al., 2016; Tomsett et al., 2020) and natural language processing (Pham et al., 2021; Kim et al., 2020; Arras et al., 2017; DeYoung et al., 2020). Deleting a pixel by zeroing it out is problematic. First, zeroing out a pixel can create OOD samples, which often cause CNNs to misbehave (Hooker et al., 2019; Agarwal & Nguyen, 2020). Second, zeroing out a pixel can generate an image of different, unintended meanings to CNNs (e.g. zero pixels represent darkness in a matchstick image), potentially leading to evaluation scores inconsistent with human notions of interpretability (Jacovi & Goldberg, 2021; Hooker et al., 2019).

Score-based metrics are strongly correlated with the average probability scores of a CNN, rendering the comparison between vanilla and robust models under Deletion and Insertion extremely biased and conclusions misleading. To the best of our knowledge, this finding is novel and different from a prior OOD-ness issue of Insertion and Deletion samples raised by Hooker et al. (2019).

As Gevaert et al. (2022); Tomsett et al. (2020), we find that Insertion and Deletion scores are highly correlated. Our results find localization-based scores (WSL and PG) to *not strongly correlate* with these two score-based metrics and therefore measure different characteristics of AMs.

## 5 Limitations

Our work is the first large-scale evaluation of 12 CNNs and nine methods, in which we cover three main sets of representative methods: gradient-based, perturbation-based, and CAM-based. We focus on CNNs for image classification. However, our methodology could be expanded to other domains, such as language models and Transformers, which are beyond the scope of this paper.

For a fair comparison, we tune the hyperparameters of each AM method for each CNN separately using a coarse grid search (Sec. A.1). That is, we only sweep across a limited range of values heuristically chosen from multiple rounds of tests. We chose to include WSL, Pointing Game, Deletion and Insertion, which are representative of localization-based and score-based metrics. However, some explainability conclusions may change when tested on ROAR (Hooker et al., 2019), which we did not include due to its excessive computational cost at the scale of our study (i.e., testing over nine AM methods, 12 CNNs, and 2,000 images). Furthermore, like any automatic evaluation metric, ROAR has its own limitations, and may or may not translate into improvement in the performance of humans (i.e. the target users of attribution maps). We leave ROAR for future work.

**Feature-attribution evaluation** The surprising results in Secs. 3.6 and 3.7 show that how to *automatically* evaluate feature attribution maps is still an open problem and despite being popular, PG, WSL, Insert, and Deletion may measure an undesired property of the network.

We acknowledge that one of the most *objective* evaluation metrics for attribution maps should be measuring how attribution maps improve human performance (e.g. efficiency or accuracy) on a downstream task (Nguyen et al., 2021). Yet, performing a human study for each triplet of (training algorithm, network architecture, and attribution method) in our large-scale study is prohibitively expensive and, therefore, out of the scope of this paper. Instead, we hope to shed light on the model selection based on the automatic evaluation metrics (PG, WSL, Insertion, and Deletion).

## 6 Conclusion

We show that robust models under gradient-based methods are significantly and consistently more explainable compared to their vanilla counterparts; however, such patterns were not seen in other categories of AMs, i.e., perturbation-based and CAM-based. Analyzing AdvProp models showed that even though they achieve higher in and out of distribution accuracies, they could not outperform adversarially robust trained models in terms of explainability. Evaluating attribution methods across all 12 CNNs revealed that GradCAM, although introduced in 2017, still seems to be the best overall AM method taking into account both run-time and overall performance of the methods across 12 networks and four metrics which somehow coincides with the results in (Choe et al., 2020). Also, it should be noted that CAM-based methods suggest almost no difference in explainability ability between robust and vanilla models which suggests that choosing the CAM-based methods is a safe choice in both scenarios.

Furthermore, interestingly, PGD-1 models perform roughly the best under PG, WSL, and Insertion on ImageNet images (Fig. 16). The overall promising results of Advprop invite future research on how to leverage AdvProp models for the actual downstream human-in-the-loop image classification tasks.

Overall, this study shed light on the strengths, similarities, and limitations of different vanilla and robust CNNs across various architectures and attribution methods, emphasizing the need to strike a balance between model accuracy and explainability. The findings provide valuable insights and recommendations for researchers and practitioners working on explainable AI on how to use attribution methods better and efficiently, which encourages further exploration and advancements in this field.

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

# A   Appendix

## A.1   Hyperparameter settings

### A.1.1   Hyperparameter tuning

Most gradient-based and CAM-based methods do not have hyperparameters while perturbation-based methods are often the most beneficial from hyperparameter tuning (Table 3).

Several of the attribution methods we consider here have hyperparameters that affect their behavior. For **IG**, Captum (Kokhlikyan et al., 2020) suggests $N_{steps} = 50$ and for **SHAP** (Lundberg & Lee, 2017b) the default is $N_{samples} = 200$. The defaults for **MP** (Gildenblat, 2021) are $\beta = 3$, learning rate $\gamma = 0.1$ , 500 iterations, $\lambda_1 = 0.01$, and $\lambda_2 = 0.2$. For **EP** (Fong & Vedaldi, 2019), default hyperparameters are areas = $[0.025, 0.05, 0.1, 0.2]$, 800 iterations, and smoothing = 0.09. For **RISE** (Petsiuk et al., 2018), $N_{masks} = 8000$ and mask size = 7. And for **LIME** (Ribeiro et al., 2016), $N_{samples} = 1000$, $N_{superpixels} = 50$.

The default or suggested hyperparameters in the literature were tuned for vanilla CNNs; the optimal values are not necessarily the same for robust CNNs. We tuned the hyperparameters for perturbation-based methods across all CNNs. We use 250 randomly chosen images from ImageNet (Russakovsky et al., 2015) that are not in ImageNet-CL. Then we define the search space as mentioned in the following paragraph and perform a grid search over all these hyperparameters for the best performance.

**EP**   We sweep across the following lists of "area sizes" : { [0.05, 0.1], [0.02, 0.08, 0.16], [0.05, 0.15, 0.3], [0.025, 0.05, 0.1, 0.2], [0.012, 0.025, 0.05, 0.1, 0.2], [0.012, 0.025, 0.05, 0.1, 0.2, 0.4],[0.015, 0.03, 0.05, 0.1, 0.25, 0.5], [0.01, 0.04, 0.05, 0.1, 0.2, 0.4] }.

**MP**   We sweep across the following numbers of steps: { 5, 10, 25, 50, 100, 200, 300, 400, 500 }.

**RISE**   We sweep across different numbers of masks: { 100, 200, 400, 600, 800, 1000, 2000, 3000, 4000, 5000, 6000, 7000, 8000, 9000, 10000 } and the different values for the mask size: {3, 4, 5, 6, 7, 8, 9}.

**LIME**   We sweep across the following values for the numbers of superpixels: {10, 20, 30, 40, 50, 60 }. Changing the number of samples shows no significant effect on LIME explanation accuracy, so we do not tune this hyperparameter.

Table 3: Best hyperparameters for each architecture and method. **Bold** numbers are the hyperparameter values found that are better than the default settings.

|  | EP (Fong et al., 2019) Area sizes | MP (Fong & Vedaldi, 2017) Number of steps | RISE (Petsiuk et al., 2018) Mask size, number of samples | LIME (Ribeiro et al., 2016) Number of superpixels |
|---|---|---|---|---|
| *Default* | [0.025, 0.05, 0.1, 0.2] | 500 | 7, 8000 | 50 |
| AlexNet | **[0.012, 0.025, 0.05, 0.1, 0.2]** | 500 | 7, 8000 | **40** |
| AlexNet-R | **[0.015, 0.03, 0.05, 0.1, 0.25, 0.5]** | **10** | **3, 4000** | **20** |
| GoogLeNet | [0.025, 0.05, 0.1, 0.2] | 500 | 7, 8000 | **40** |
| GoogLeNet-R | [0.025, 0.05, 0.1, 0.2] | **10** | **3, 8000** | **40** |
| ResNet | [0.025, 0.05, 0.1, 0.2] | 500 | 7, 8000 | 50 |
| ResNet-R | [0.025, 0.05, 0.1, 0.2] | **10** | **4, 7000** | **20** |
| MobileNet | [0.025, 0.05, 0.1, 0.2] | **400** | **6, 7000** | **40** |
| MobileNet-R | [0.025, 0.05, 0.1, 0.2] | **10** | **4, 8000** | **20** |
| DenseNet | [0.025, 0.05, 0.1, 0.2] | 500 | 7, 8000 | 50 |
| DenseNet-R | [0.025, 0.05, 0.1, 0.2] | **10** | **4, 7000** | 50 |
| PGD-5 | [0.025, 0.05, 0.1, 0.2] | 500 | 7, 8000 | 50 |
| PGD-1 | [0.025, 0.05, 0.1, 0.2] | 500 | 7, 8000 | **40** |

### A.1.2   Best hyperparameter settings

As the result of tuning, below are the final hyperparameter settings for each method used in the paper.

**EP**        Implementation:        `https://facebookresearch.github.io/TorchRay/attribution.html#module-torchray.attribution.extremal_perturbation`

- Iterations = 800
- Smoothing = 0.09
- reward function = simple reward
- Perturbation = 'blur'
- Step = 7
- Sigma = 21
- Jitter = True
- Areas sizes : see Table 3

**MP** Implementation: `https://github.com/jacobgil/pytorch-explain-black-box`

- $\beta = 3$
- learning rate $\gamma = 0.1$
- Iterations = 500
- $\lambda_1 = 0.01$
- $\lambda_2 = 0.2$
- Number of steps : see Table 3

**RISE** Implementation: `https://github.com/eclique/RISE`

- probability of producing 1s in the binary masks = 0.5
- Mask and Number of samples : see Table 3

**LIME** Implementation: `https://github.com/marcotcr/lime/blob/master/doc/notebooks/Tutorial%20-%20images%20-%20Pytorch.ipynb`

- Number of samples = 1000
- Segmenter parameters:
      Compactness = 10
      Sigma = 3
- Number of superpixels : see Table 3

**SHAP** Implementation: `https://shap-lrjball.readthedocs.io/en/latest/generated/shap.GradientExplainer.html`

- Number of samples = 200

**IG** Implementation: `https://captum.ai/api/integrated_gradients.html`

- Number of steps = 50

**Gradient** Implementation: `https://facebookresearch.github.io/TorchRay/attribution.html#module-torchray.attribution.gradient`

**GradCAM** Implementation: `https://github.com/jacobgil/pytorch-grad-cam`

**CAM** Implementation: `https://github.com/zhoubolei/CAM`

### A.2  ImageNet-CL results continuation

### A.2.1  CNNs ranking for Pointing Game and Deletion

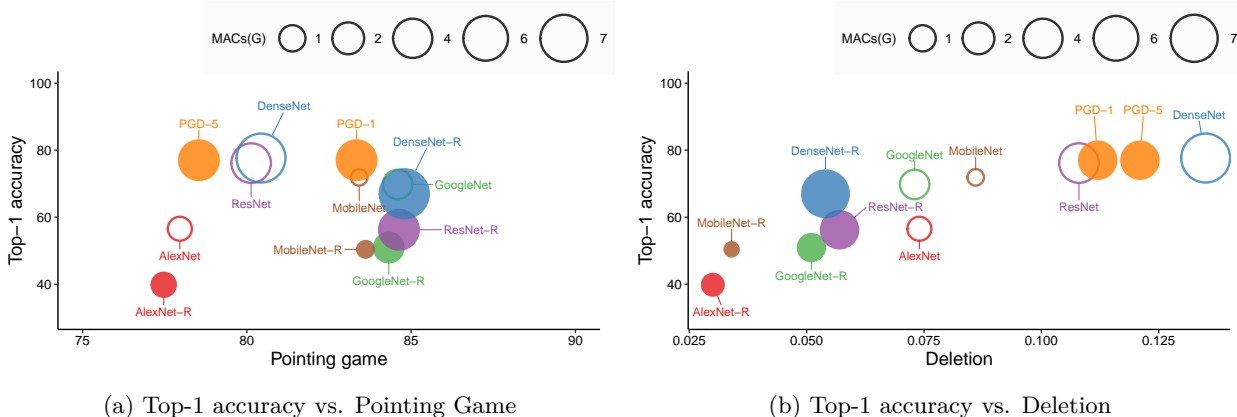

(a) Top-1 accuracy vs. Pointing Game   (b) Top-1 accuracy vs. Deletion

Figure 9: Shows the average performance of all CNNs across eight attribution methods for Pointing Game and Deletion along with top-1 accuracy and MACs. See Fig. 16 for the same plot for WSL and Insertion.

### A.2.2 Insertion and Deletion examples to show the bias of these two metrics

This section is an extension for Fig. 8a to demonstrate for the bias of Insertion and Deletion.

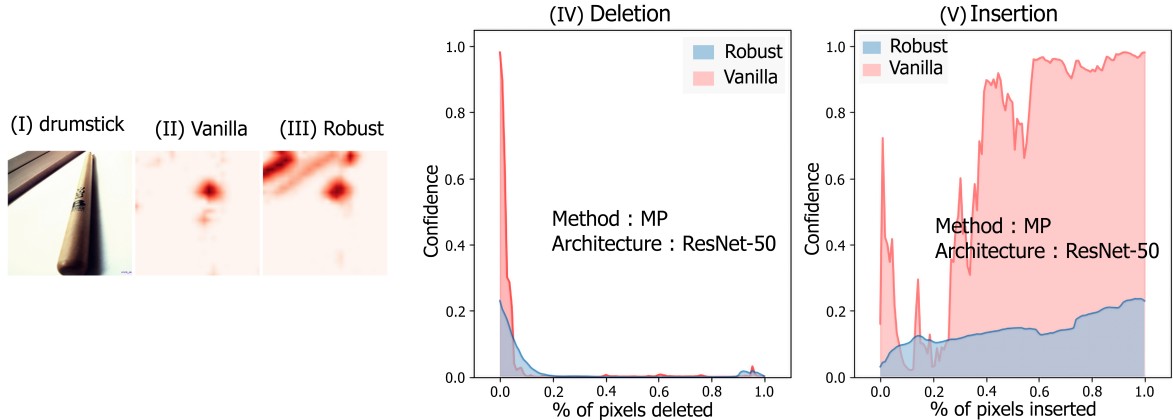

Figure 10: Given the same "drumstick" image (I), the MP heatmaps of ResNet and ResNet-R are not of good quality. Yet, under Deletion (robust AUC = 0.01 vs. vanilla AUC=0.03), the robust heatmap (III) is better. In contrast, under Insertion, the vanilla heatmap (II) is better.

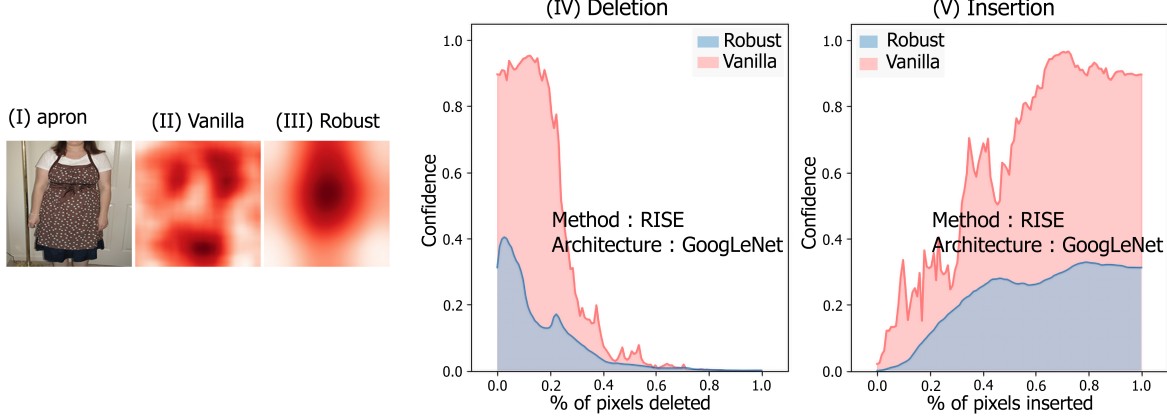

Figure 11: Given the same "apron" image (I), the RISE heatmaps of GoogleNet and GooleNet-R are shown in (II) and (III), respectively. Again as we saw in Fig. 10, under Deletion, the robust heatmap (III) is better. In contrast, under Insertion, the vanilla heatmap (II) is better.

## A.3 Rankings of methods, architectures and networks

Table 4: Mean ± std and ranking of AM methods as shown in Fig. 4 over all CNNs shown in bold on **ImageNet-CL**. Average of rankings is the mean value of rankings across the evaluation metrics.

| | Pointing game | WSL | Insertion | Deletion | Average of rankings |
|---|---|---|---|---|---|
| **GradCAM** | 85.32 ± 10.19 (**3rd**) | 62.29 ± 14.17 (**1st**) | 0.57 ± 0.19 (**2nd**) | 0.07 ± 0.03 (**3rd**) | 2.25 |
| **RISE** | 91.41 ± 1.87 (**1st**) | 56.72 ± 7.03 (**3rd**) | 0.57 ± 0.19 (**2nd**) | 0.07 ± 0.03 (**3rd**) | 2.25 |
| **EP** | 89.70 ± 2.07 (**2nd**) | 59.06 ± 2.53 (**2nd**) | 0.59 ± 0.19 (**1st**) | 0.08 ± 0.04 (**6th**) | 2.75 |
| **IG** | 82.24 ± 4.84 (**4th**) | 44.97 ± 4.23 (**6th**) | 0.53 ± 0.12 (**6th**) | 0.06 ± 0.03 (**2nd**) | 4.5 |
| **SHAP** | 81.91 ± 4.81 (**5th**) | 42.61 ± 6.63 (**7th**) | 0.53 ± 0.12 (**6th**) | 0.05 ± 0.03 (**1st**) | 4.75 |
| **LIME** | 78.01 ± 4.36 (**7th**) | 45.47 ± 2.91 (**5th**) | 0.56 ± 0.19 (**4th**) | 0.07 ± 0.03 (**3rd**) | 4.75 |
| **Gradient** | 78.36 ± 7.81 (**6th**) | 47.03 ± 6.88 (**4th**) | 0.50 ± 0.11 (**8th**) | 0.08 ± 0.04 (**6th**) | 6 |
| **MP** | 70.14 ± 3.51 (**8th**) | 31.58 ± 3.12 (**8th**) | 0.56 ± 0.14 (**4th**) | 0.08 ± 0.03 (**6th**) | 6.5 |

Table 5: Mean ± std and ranking of AM methods as shown in Fig. 15 over all CNNs shown in bold on **ImageNet**. Average of rankings is the mean value of rankings across our four evaluation metrics.

| | Pointing game | WSL | Insertion | Deletion | Average of rankings |
|---|---|---|---|---|---|
| **EP** | 84.63 ± 4.04 (**1st**) | 47.42 ± 3.16 (**2nd**) | 0.34 ± 0.17 (**2nd**) | 0.04 ± 0.02 (**3rd**) | 2 |
| **RISE** | 82.20 ± 7.65 (**2nd**) | 42.62 ± 5.95 (**3rd**) | 0.36 ± 0.18 (**1st**) | 0.04 ± 0.02 (**3rd**) | 2.25 |
| **GradCAM** | 77.95 ± 10.35 (**3rd**) | 51.63 ± 11.82 (**1st**) | 0.33 ± 0.17 (**3rd**) | 0.04 ± 0.02 (**3rd**) | 2.5 |
| **IG** | 77.14 ± 4.14 (**4th**) | 38.34 ± 3.71 (**5th**) | 0.30 ± 0.12 (**6th**) | 0.03 ± 0.02 (**1st**) | 4 |
| **SHAP** | 76.85 ± 3.80 (**5th**) | 36.43 ± 5.31 (**6th**) | 0.30 ± 0.12 (**6th**) | 0.03 ± 0.02 (**1st**) | 4.5 |
| **LIME** | 67.07 ± 7.64 (**7th**) | 35.07 ± 4.93 (**7th**) | 0.33 ± 0.17 (**3rd**) | 0.04 ± 0.02 (**3rd**) | 5 |
| **Gradient** | 74.04 ± 7.09 (**6th**) | 40.43 ± 5.30 (**4th**) | 0.28 ± 0.11 (**8th**) | 0.05 ± 0.03 (**7th**) | 6.25 |
| **MP** | 65.61 ± 5.64 (**8th**) | 25.72 ± 3.93 (**8th**) | 0.33 ± 0.15 (**3rd**) | 0.05 ± 0.02 (**7th**) | 6.5 |

Table 6: Ranking of CNNs considering under all evaluation metrics, top-1 accuracy, and MACs on **ImageNet-CL**.

| | Pointing game | WSL | Insertion | Deletion | Top-1 accuracy | MACs (G) |
|---|---|---|---|---|---|---|
| **AlexNet** | 77.96 (**11th**) | 43.81 (**11th**) | 0.582 (**7th**) | 0.074 (**7th**) | 56.55 (**8th**) | 0.72 (**3rd**) |
| **AlexNet-R** | 77.47 (**12th**) | 41.78 (**12th**) | 0.333 (**12th**) | 0.03 (**1st**) | 39.83 (**12th**) | 0.72 (**3rd**) |
| **GoogleNet** | 84.59 (**3rd**) | 50.9 (**4th**) | 0.632 (**5th**) | 0.073 (**6th**) | 69.78 (**6th**) | 1.51 (**5th**) |
| **GoogleNet-R** | 84.32 (**4th**) | 52.59 (**2nd**) | 0.428 (**10th**) | 0.051 (**3rd**) | 50.94 (**10th**) | 1.51 (**5th**) |
| **DenseNet** | 80.43 (**8th**) | 45.67 (**10th**) | 0.78 (**2nd**) | 0.135 (**12th**) | 77.14 (**2nd**) | 7.82 (**11th**) |
| **DenseNet-R** | 84.79 (**1st**) | 52.06 (**3rd**) | 0.579 (**8th**) | 0.054 (**4th**) | 66.12 (**7th**) | 7.82 (**11th**) |
| **MobileNet** | 83.42 (**6th**) | 50.31 (**6th**) | 0.611 (**6th**) | 0.086 (**8th**) | 71.88 (**5th**) | 0.32 (**1st**) |
| **MobileNet-R** | 83.61 (**5th**) | 50.65 (**5th**) | 0.338 (**11th**) | 0.034 (**2nd**) | 50.4 (**11th**) | 0.32 (**1st**) |
| **ResNet** | 80.13 (**9th**) | 46.11 (**9th**) | 0.727 (**4th**) | 0.108 (**9th**) | 76.15 (**4th**) | 4.12 (**9th**) |
| **ResNet-R** | 84.63 (**2nd**) | 53.27 (**1st**) | 0.496 (**9th**) | 0.057 (**5th**) | 56.25 (**9th**) | 4.12 (**9th**) |
| **PGD-5** | 78.54 (**10th**) | 48.07 (**8th**) | 0.796 (**1st**) | 0.121 (**11th**) | 77.01 (**3rd**) | 4.1 (**7th**) |
| **PGD-1** | 83.34 (**7th**) | 50.11 (**7th**) | 0.778 (**3rd**) | 0.112 (**10th**) | 77.31 (**1st**) | 4.1 (**7th**) |

Table 7: Median (interquartile range) and architectures' rank (bold) shown in Fig. 5 for each metric on **ImageNet-CL**. Average of rankings is the mean value of rankings across the evaluation metrics.

| | Pointing Game | WSL | Insertion | Deletion | Average of rankings |
|---|---|---|---|---|---|
| **ResNet-50** | 86.40 (17.71), **2nd** | 51.75 (18.40), **1st** | 0.56 (0.26), **2nd** | 0.08 (0.06), **4th** | 2.25 |
| **GoogLeNet** | 86.13 (10.05), **3rd** | 48.88 (11.13), **2nd** | 0.50 (0.22), **3rd** | 0.06 (0.03), **3rd** | 2.75 |
| **DenseNet-161** | 86.58 (13.01), **1st** | 46.33 (15.76), **4th** | 0.64 (0.21), **1st** | 0.10 (0.08), **5th** | 2.75 |
| **MobileNet** | 83.90 (8.80), **4th** | 48.00 (12.14), **3rd** | 0.42 (0.25), **5th** | 0.06 (0.05), **2nd** | 3.5 |
| **AlexNet** | 79.90 (11.45), **5th** | 40.75 (12.25), **5th** | 0.44 (0.24), **4th** | 0.04 (0.04), **1st** | 3.75 |

### A.4   Results and analysis of ImageNet

In this section, we explain the results that have been obtained by using ImageNet defined in Sec. 2.1 in comparison with ImageNet-CL results demonstrated in Sec. 3.

#### A.4.1   Robust models outperform their vanilla counterpart in gradient-based methods

As we can see in Fig. 12, we reach the same conclusion as in Sec. 3.1 for CAM-based and Gradient-based methods as we previously saw for ImageNet-CL. The only difference is about Perturbation-based methods, in which we see vanilla models outperform robust models.

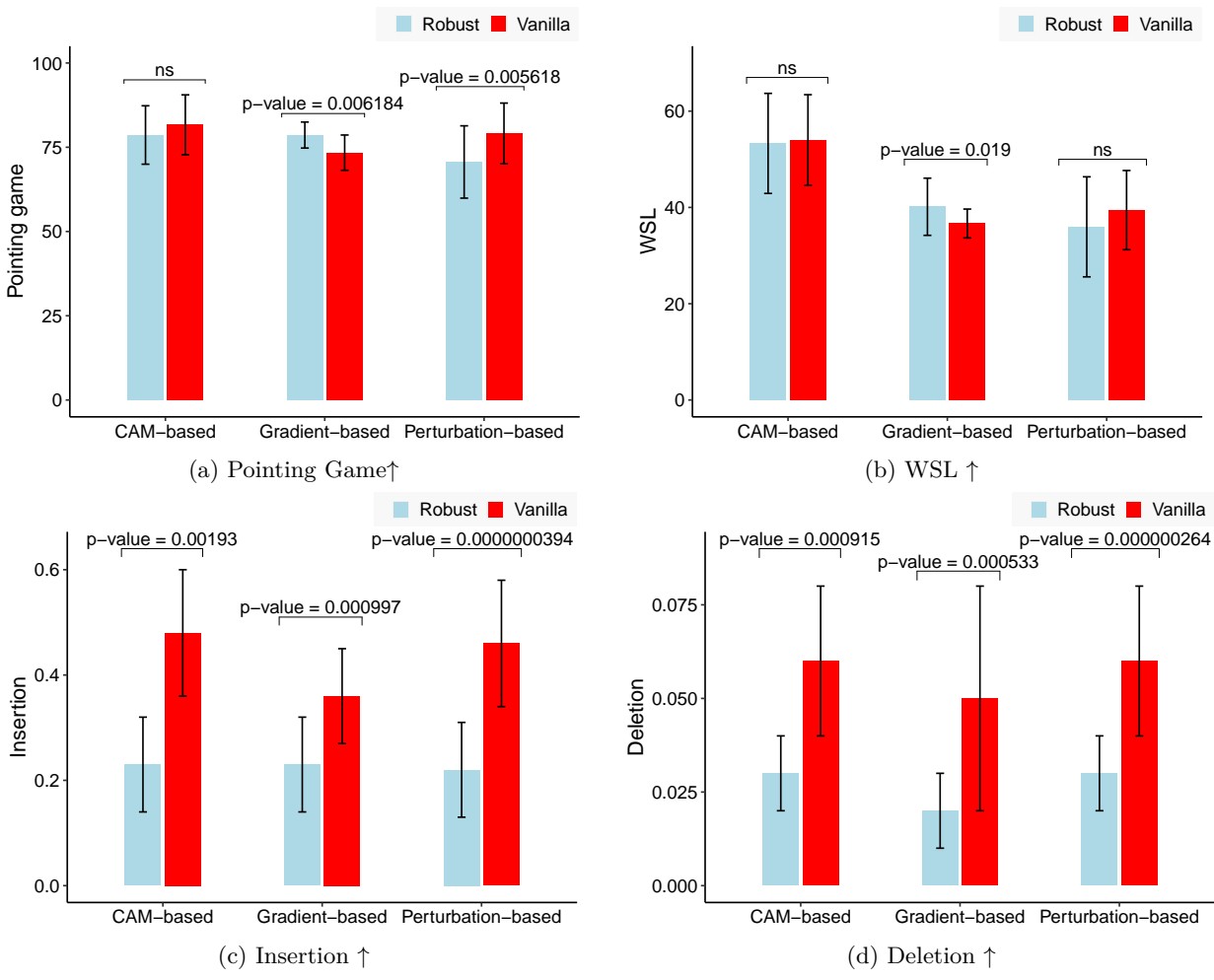

Figure 12: Performance of explanations generated from vanilla and robust models for three different types of AM methods across four different metrics on ImageNet images which are aligned with the conclusion in Sec. 3.1; Robust models outperform vanilla ones in gradient-based methods but not in CAM-based nor perturbation-based methods.

### A.4.2 AdvProp models outperform vanilla models in both classifiability and explainability

We also conclude the same result as we saw in Sec. 3.5 that an incremental trend is visible from vanilla to robust models by AdvProp models being in between, as we can see in Fig. 13 for localization metrics.

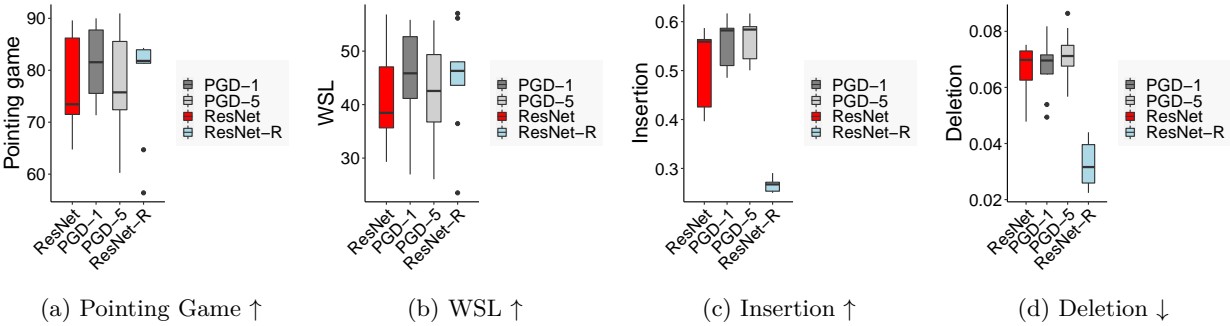

(a) Pointing Game ↑     (b) WSL ↑     (c) Insertion ↑     (d) Deletion ↓

Figure 13: AdvProp models outperform vanilla models in all metrics excpet Deletion (d), i.e. Pointing Game (a), WSL (b), and Insertion (c). Also, as shown, it supports the smooth transition we saw in ImageNet-CL (see Fig. 7).

### A.4.3 ResNet-50 and DenseNet-161 are the top architecture to use

ResNet-50 is the top architecture as shown in Fig. 14, which we saw in Sec. 3.3. MobileNet-v2 and AlexNet are the best architectures in Deletion, while they are consistently the worst for the other three evaluation metrics.

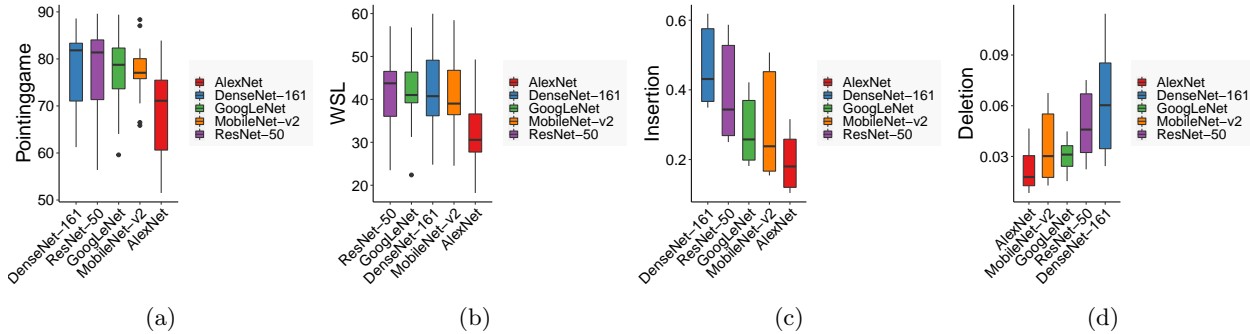

(a)     (b)     (c)     (d)

Figure 14: ResNet-50 is the overall best architecture as we concluded on ImageNet-CL before (see Sec. 3.3). Also, even though AlexNet and MobileNet-v2 are the best architectures under Deletion, they are consistently the worst across the other three metrics.

### A.4.4 EP, RISE, and GradCAM are the best AM methods considering all metrics

The three best AMs overall came out to be EP, RISE, and GradCAM on ImageNet (see Table 5 for average of rankings), as shown in Fig. 15. These top-3 methods are what we obtained for ImageNet-CL in Sec. 3.2.

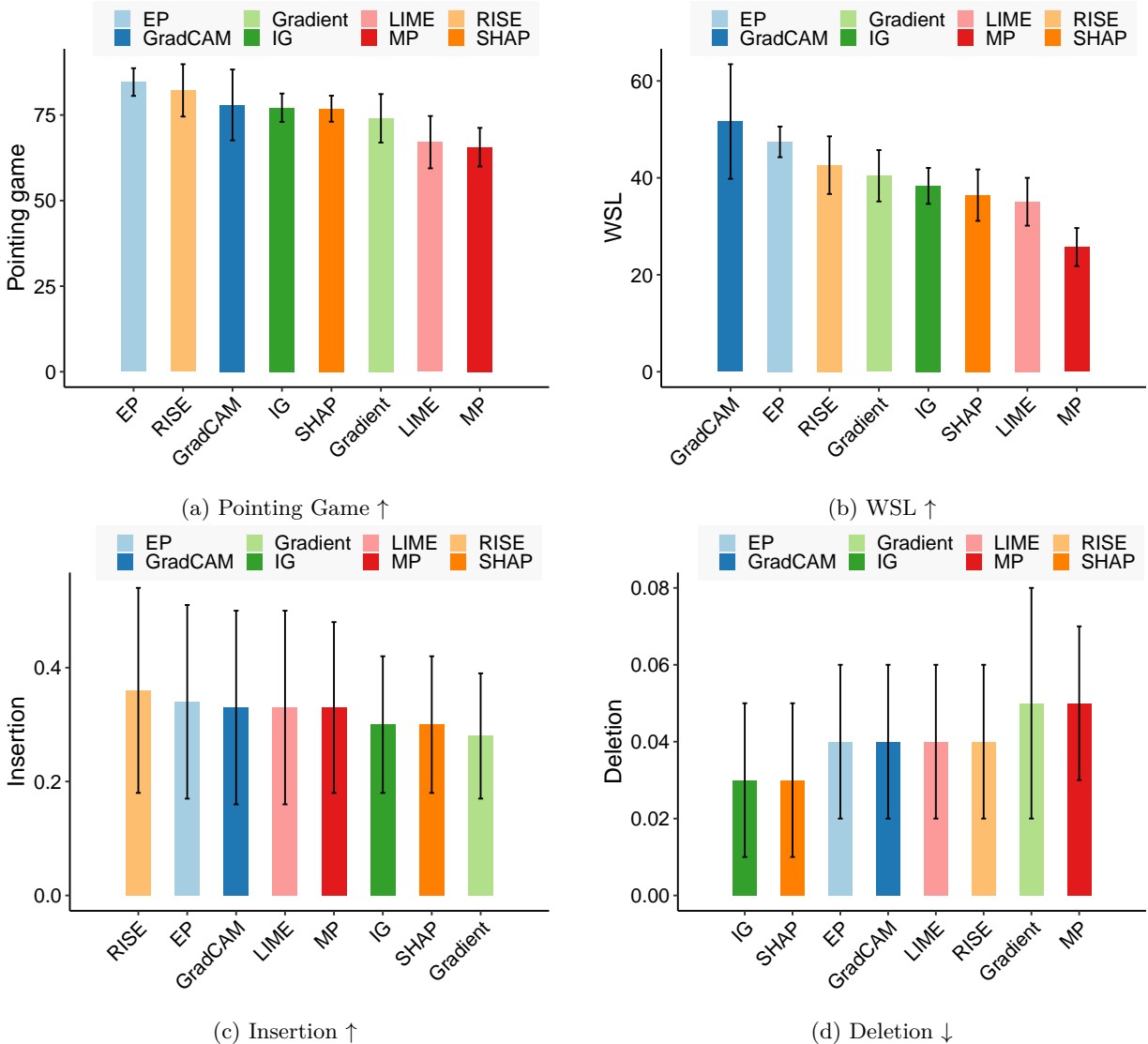

Figure 15: EP, RISE, and GradCAM are the best AM methods as they are consistently among top-3 for all metrics, which we concluded on ImageNet-CL (see Sec. 3.2) as well. Moreover, MP is the worst method as it the last in the ranking in all metrics except Insertion (c) which is exactly the same on ImageNet-CL (see Fig. 4).

### A.4.5 CNN classification accuracy, explanations, and number of multiply–accumulate operations

As explained in Sec. 3.4 we find no CNNs to be the best across all evaluation metrics and criteria. Considering explanation accuracy, top-1 accuracy, and MACs, PGD-1 model is the best, as depicted in Fig. 16 on average. This is the same as what we concluded for ImageNet-CL (see Fig. 6).

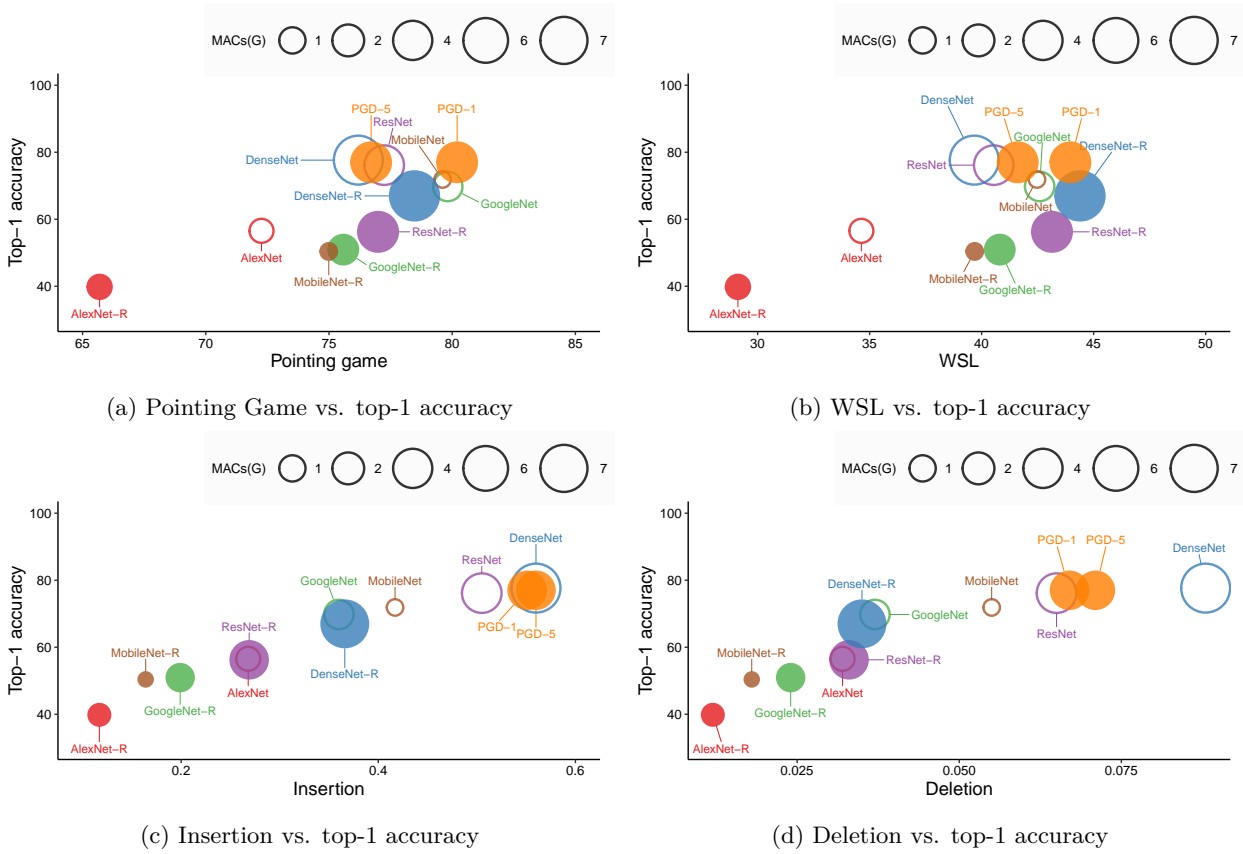

(a) Pointing Game vs. top-1 accuracy

(b) WSL vs. top-1 accuracy

(c) Insertion vs. top-1 accuracy

(d) Deletion vs. top-1 accuracy

Figure 16: The average explainability score vs. MACs and top-1 accuracy show that no CNN is the best across all metrics and criteria as we stated in Sec. 3.4. On average, it seems that AdvProp model, i.e. PGD-1, is the best on average since it is among the best for Insertion, WSL, Pointing Game, and top-1 accuracy.

## A.5 Quantitative results of ImageNet-CL

In this section, the quantitative results of the experiment in Sec. 3.1 will be presented for the evaluation metrics for all CNNs and AM methods on ImageNet-CL.

Table 8: Pointing game for all 9 AM methods and 12 CNNS for experiment in Sec. 3.1 on **ImageNet-CL**. Bold numbers are the best CNNs for each AM method. In gradient-based methods, robust models are performing better by a great deal compared to their vanilla counter parts as our results in Sec. 3.1 suggests.

|  | Gradient-based | | | CAM-based | | Perturbation-based | | | |
|---|---|---|---|---|---|---|---|---|---|
|  | Gradient | IG | SHAP | GradCAM | CAM | RISE | LIME | MP | EP |
| AlexNet | 76.50 | 81.55 | 80.70 | 64.45 | N/A | 88.55 | 71.35 | 71.60 | 88.95 |
| AlexNet-R | 79.85 | 81.30 | 79.95 | 68.25 | N/A | 87.60 | 69.45 | 68.70 | 84.65 |
| GoogLeNet | 76.40 | 83.15 | 83.20 | **92.65** | **92.65** | 93.00 | 82.10 | **74.15** | **92.10** |
| GoogLeNet-R | 85.75 | 86.75 | 86.50 | 90.85 | 90.85 | 92.55 | 78.15 | 64.60 | 89.40 |
| ResNet | 64.50 | 72.30 | 74.05 | 91.95 | 92.00 | **93.20** | 81.05 | 72.70 | 91.25 |
| ResNet-R | **86.15** | 86.65 | **87.35** | 91.95 | 91.95 | 92.05 | 78.80 | 63.95 | 90.15 |
| MobileNet | 79.05 | 83.05 | 79.90 | 88.40 | N/A | 92.10 | 81.20 | 72.45 | 91.20 |
| MobileNet-R | 83.25 | 84.55 | 85.05 | 87.95 | N/A | 91.00 | 78.50 | 69.85 | 88.75 |
| DenseNet | 66.15 | 75.95 | 75.10 | 88.35 | 88.10 | 91.95 | 82.00 | 73.20 | 90.70 |
| DenseNet-R | 86.00 | **87.15** | 87.25 | 88.40 | 85.30 | 92.05 | 77.50 | 70.15 | 89.80 |
| PGD-5 | 73.15 | 82.75 | 77.10 | 58.80 | 91.10 | 92.15 | **82.50** | 70.65 | 91.20 |
| PGD-1 | 75.60 | 84.10 | 79.4 | 89.50 | 92.05 | 92.75 | 82.15 | 72.10 | 91.15 |

Table 9: WSL for all 9 AM methods and 12 CNNS for experiment in Sec. 3.1 on **ImageNet-CL**. Bold numbers are the best CNNs for each AM method. Mostly, robust CNNs achieving the maximum score (6 out 9 AM methods).

|  | Gradient-based | | | CAM-based | | Perturbation-based | | | |
|---|---|---|---|---|---|---|---|---|---|
|  | Gradient | IG | SHAP | GradCAM | CAM | RISE | LIME | MP | EP |
| AlexNet | 42.95 | 43.90 | 39.95 | 35.30 | N/A | 52.95 | 41.55 | 30.70 | 63.15 |
| AlexNet-R | 46.25 | 36.70 | 32.85 | 35.80 | N/A | 57.35 | 39.80 | 26.70 | 58.80 |
| GoogLeNet | 48.30 | 47.45 | 45.05 | 68.65 | 68.70 | 49.25 | 48.65 | **37.85** | 62.00 |
| GoogLeNet-R | 52.50 | 48.60 | 49.10 | 69.35 | **69.50** | **67.45** | 46.75 | 29.75 | 57.20 |
| ResNet | 36.45 | 40.55 | 36.25 | 66.10 | 65.80 | 52.60 | 45.50 | 33.45 | 58.00 |
| ResNet-R | 54.05 | 51.35 | **52.15** | 67.15 | 67.10 | 65.75 | 46.95 | 30.85 | 57.90 |
| MobileNet | 47.25 | 46.25 | 38.85 | **71.05** | N/A | 53.45 | **48.75** | 34.25 | 62.65 |
| MobileNet-R | 50.55 | 46.10 | 45.40 | 69.15 | N/A | 60.65 | 47.20 | 28.95 | 57.20 |
| DenseNet | 35.95 | 42.35 | 36.5 | 69.45 | 65.70 | 46.20 | 44.90 | 32.75 | 57.25 |
| DenseNet-R | **56.05** | 46.45 | 50.00 | 70.85 | 69.00 | 61.55 | 44.60 | 30.55 | 56.45 |
| PGD-5 | 44.90 | 49.60 | 42.35 | 57.30 | 66.15 | 48.45 | 45.85 | 30.15 | **65.95** |
| PGD-1 | 48.90 | **51.80** | 45.60 | 61.10 | 66.70 | 50.50 | 45.05 | 33.25 | 64.65 |

Table 10: Insertion for all 9 AM methods and 12 CNNS for experiment in Sec. 3.1 on **ImageNet-CL**. Bold numbers are the best CNNs for each AM method. As we stated in Sec. 3.6, we can observe that two CNNs with the high mean confidence are dominating Insertion; PGD-5 and DenseNet.

| | Gradient-based | | | CAM-based | | Perturbation-based | | | |
|---|---|---|---|---|---|---|---|---|---|
| | Gradient | IG | SHAP | GradCAM | CAM | RISE | LIME | MP | EP |
| AlexNet | 0.522 | 0.553 | 0.559 | 0.579 | N/A | 0.643 | 0.593 | 0.584 | 0.627 |
| AlexNet-R | 0.347 | 0.356 | 0.342 | 0.307 | N/A | 0.311 | 0.298 | 0.361 | 0.339 |
| GoogLeNet | 0.536 | 0.575 | 0.579 | 0.670 | 0.670 | 0.708 | 0.670 | 0.637 | 0.682 |
| GoogLeNet-R | 0.431 | 0.459 | 0.465 | 0.401 | 0.401 | 0.401 | 0.397 | 0.453 | 0.419 |
| ResNet | 0.585 | 0.630 | 0.631 | 0.800 | 0.799 | 0.819 | 0.803 | 0.738 | 0.809 |
| ResNet-R | 0.500 | 0.530 | 0.536 | 0.466 | 0.466 | 0.469 | 0.471 | 0.517 | 0.481 |
| MobileNet | 0.490 | 0.537 | 0.514 | 0.742 | N/A | 0.641 | 0.628 | 0.579 | 0.759 |
| MobileNet-R | 0.343 | 0.355 | 0.353 | 0.319 | N/A | 0.324 | 0.320 | 0.358 | 0.334 |
| DenseNet | 0.682 | 0.719 | 0.723 | 0.822 | 0.823 | 0.842 | 0.832 | **0.789** | **0.832** |
| DenseNet-R | 0.589 | 0.604 | 0.602 | 0.552 | 0.545 | 0.556 | 0.556 | 0.597 | 0.577 |
| PGD-5 | **0.733** | **0.764** | **0.752** | **0.825** | **0.826** | **0.847** | **0.835** | 0.786 | 0.830 |
| PGD-1 | 0.699 | 0.733 | 0.716 | 0.820 | 0.818 | 0.839 | 0.827 | 0.773 | 0.820 |

Table 11: Deletion for all 9 AM methods and 12 CNNS for experiment in Sec. 3.1 on **ImageNet-CL**. Bold numbers are the best CNNs for each AM method. As we stated in Sec. 3.6, we can observe that two CNNs with the lowest mean confidence are dominating Deletion; PGD-5 and DenseNet. The only case that none of them are winner is CAM in which we did not have MobileNet-v2 and AlexNet architecture.

| | Gradient-based | | | CAM-based | | Perturbation-based | | | |
|---|---|---|---|---|---|---|---|---|---|
| | Gradient | IG | SHAP | GradCAM | CAM | RISE | LIME | MP | EP |
| AlexNet | 0.061 | 0.042 | 0.039 | 0.093 | N/A | 0.070 | 0.111 | 0.094 | 0.084 |
| AlexNet-R | **0.033** | **0.022** | **0.020** | **0.034** | N/A | 0.030 | **0.029** | 0.044 | **0.032** |
| GoogLeNet | 0.089 | 0.064 | 0.060 | 0.070 | 0.070 | 0.067 | 0.086 | 0.079 | 0.075 |
| GoogLeNet-R | 0.064 | 0.036 | 0.032 | 0.048 | **0.049** | 0.042 | 0.063 | 0.074 | 0.049 |
| ResNet | 0.125 | 0.086 | 0.077 | 0.115 | 0.115 | 0.108 | 0.108 | 0.125 | 0.122 |
| ResNet-R | 0.078 | 0.045 | 0.039 | 0.056 | 0.056 | 0.050 | 0.042 | 0.087 | 0.057 |
| MobileNet | 0.091 | 0.069 | 0.068 | 0.109 | N/A | 0.073 | 0.079 | 0.085 | 0.112 |
| MobileNet-R | 0.043 | 0.032 | 0.024 | 0.034 | N/A | **0.029** | 0.032 | **0.040** | 0.035 |
| DenseNet | 0.174 | 0.132 | 0.124 | 0.119 | 0.119 | 0.134 | 0.120 | 0.144 | 0.130 |
| DenseNet-R | 0.065 | 0.048 | 0.038 | 0.054 | 0.057 | 0.048 | 0.051 | 0.074 | 0.056 |
| PGD-5 | 0.140 | 0.095 | 0.108 | 0.119 | 0.119 | 0.117 | 0.121 | 0.154 | 0.118 |
| PGD-1 | 0.120 | 0.083 | 0.091 | 0.115 | 0.119 | 0.112 | 0.121 | 0.139 | 0.119 |

## A.6 Quantitative results of ImageNet

In this section, the quantitative results of experiment in Sec. 3.1 would be presented for our evaluation metrics for all CNNs and AM methods on ImageNet.

Table 12: Pointing game for all 9 AM methods and 12 CNNS for experiment in Sec. 3.1 on **ImageNet**. Bold numbers are the best CNNs for each AM method. In gradient-based methods, DenseNet-R is dominantly the best, and AdvProp's CNNs in CAM-based methods outperform other CNNs.

|  | Gradient-based | | | CAM-based | | Perturbation-based | | | |
|---|---|---|---|---|---|---|---|---|---|
|  | Gradient | IG | SHAP | GradCAM | CAM | RISE | LIME | MP | EP |
| AlexNet | 70.60 | 76.55 | 75.70 | 60.10 | N/A | 82.25 | 60.80 | 68.20 | 83.90 |
| AlexNet-R | 71.60 | 72.30 | 72.75 | 57.90 | N/A | 64.65 | 51.50 | 59.45 | 75.40 |
| GoogleNet | 74.25 | 79.55 | 78.90 | **86.35** | 86.35 | 87.35 | 71.85 | 70.90 | **89.40** |
| GoogleNet-R | 78.50 | 78.60 | 77.95 | 82.10 | 82.10 | 80.90 | 64.05 | 59.60 | 82.95 |
| ResNet | 64.75 | 70.90 | 73.45 | 86.15 | 86.20 | 89.60 | 73.45 | **71.50** | 88.10 |
| ResNet-R | 81.80 | 81.35 | 81.40 | 83.95 | 83.95 | 82.05 | 64.70 | 56.40 | 84.30 |
| MobileNet | 76.15 | 79.45 | 76.65 | 82.20 | N/A | 88.35 | **76.50** | 70.55 | 87.05 |
| MobileNet-R | 78.35 | 78.50 | 77.25 | 76.85 | N/A | 74.75 | 66.50 | 65.85 | 81.85 |
| DenseNet | 61.25 | 71.70 | 71.15 | 82.75 | 83.15 | 88.60 | 75.70 | 70.75 | 87.60 |
| DenseNet-R | **83.10** | **82.50** | **83.30** | 81.15 | 81.15 | 83.45 | 65.65 | 62.85 | 85.75 |
| PGD-5 | 72.40 | 81.55 | 75.75 | 60.25 | 85.55 | **90.95** | 75.15 | 68.65 | 88.95 |
| PGD-1 | 74.20 | 81.55 | 75.55 | 85.30 | **87.75** | 90.00 | 75.75 | 71.35 | 87.90 |

Table 13: WSL for all 9 AM methods and 12 CNNS for experiment in Sec. 3.1 on **ImageNet**. Bold numbers are the best CNNs for each AM method. As shown in Table 9, mostly robust CNNs are achieving the maximum score (6 out 9 AM methods).

|  | Gradient-based | | | CAM-based | | Perturbation-based | | | |
|---|---|---|---|---|---|---|---|---|---|
|  | Gradient | IG | SHAP | GradCAM | CAM | RISE | LIME | MP | EP |
| AlexNet | 34.40 | 36.95 | 30.85 | 30.80 | N/A | 39.40 | 29.80 | 25.55 | 49.30 |
| AlexNet-R | 36.50 | 30.35 | 26.95 | 28.00 | N/A | 29.65 | 23.85 | 18.20 | 39.45 |
| GoogleNet | 40.90 | 41.15 | 38.80 | 56.75 | 56.35 | 43.20 | **39.35** | **31.25** | 49.35 |
| GoogleNet-R | 42.90 | 39.85 | 39.75 | 55.05 | 55.15 | 47.60 | 32.95 | 22.40 | 45.95 |
| ResNet | 35.55 | 36.20 | 35.65 | 56.55 | 56.85 | 45.55 | 38.45 | 29.30 | 47.05 |
| ResNet-R | 46.30 | 43.90 | 43.60 | 57.05 | 56.15 | 46.35 | 36.45 | 23.50 | 48.00 |
| MobileNet | 40.55 | 39.05 | 34.95 | 58.45 | N/A | 47.45 | 39.00 | 29.65 | 50.65 |
| MobileNet-R | 43.35 | 37.10 | 36.65 | 55.65 | N/A | 37.90 | 35.80 | 24.55 | 46.55 |
| DenseNet | 34.05 | 37.40 | 33.40 | 58.05 | 58.25 | 40.05 | 36.90 | 28.00 | 49.50 |
| DenseNet-R | **49.75** | 41.45 | **43.70** | **59.95** | **59.40** | **49.00** | 38.10 | 24.80 | 48.40 |
| PGD-5 | 42.40 | 44.40 | 36.45 | 49.35 | 55.75 | 42.55 | 36.75 | 26.05 | **54.95** |
| PGD-1 | 44.95 | **47.20** | 41.15 | 52.70 | 55.85 | 45.85 | 38.55 | 26.95 | 54.30 |

Table 14: Insertion for all 9 AM methods and 12 CNNS for experiment in Sec. 3.1 on **ImageNet**. Bold numbers are the best CNNs for each AM method. As shown, CNNs with high mean confidence tend to have higher higher Insertion.

| | Gradient-based | | | CAM-based | | Perturbation-based | | | |
|---|---|---|---|---|---|---|---|---|---|
| | **Gradient** | **IG** | **SHAP** | **GradCAM** | **CAM** | **RISE** | **LIME** | **MP** | **EP** |
| **AlexNet** | 0.235 | 0.248 | 0.252 | 0.280 | N/A | 0.316 | 0.253 | 0.276 | 0.284 |
| **AlexNet-R** | 0.122 | 0.121 | 0.121 | 0.112 | N/A | 0.114 | 0.104 | 0.125 | 0.118 |
| **GoogleNet** | 0.300 | 0.319 | 0.324 | 0.384 | 0.384 | 0.422 | 0.378 | 0.367 | 0.390 |
| **GoogleNet-R** | 0.198 | 0.212 | 0.216 | 0.187 | 0.186 | 0.199 | 0.182 | 0.204 | 0.195 |
| **ResNet** | 0.397 | 0.425 | 0.426 | 0.559 | 0.559 | 0.587 | 0.564 | 0.517 | 0.564 |
| **ResNet-R** | 0.269 | 0.287 | 0.291 | 0.253 | 0.254 | 0.267 | 0.250 | 0.272 | 0.263 |
| **MobileNet** | 0.304 | 0.331 | 0.318 | 0.470 | N/A | 0.507 | 0.484 | 0.446 | 0.479 |
| **MobileNet-R** | 0.168 | 0.172 | 0.172 | 0.156 | N/A | 0.161 | 0.154 | 0.169 | 0.164 |
| **DenseNet** | 0.482 | 0.506 | 0.510 | **0.590** | **0.592** | **0.619** | **0.605** | **0.571** | **0.598** |
| **DenseNet-R** | 0.370 | 0.380 | 0.380 | 0.349 | 0.350 | 0.361 | 0.353 | 0.367 | 0.366 |
| **PGD-5** | **0.501** | **0.524** | **0.513** | **0.590** | 0.587 | 0.617 | 0.594 | 0.553 | 0.584 |
| **PGD-1** | 0.485 | 0.510 | 0.497 | 0.588 | 0.587 | 0.617 | 0.583 | 0.548 | 0.582 |

Table 15: Deletion for all 9 AM methods and 12 CNNS for experiment in Sec. 3.1 on **ImageNet**. Bold numbers are the best CNNs for each AM method. The same trend in Table 11 can be seen here that AlexNet-R is dominating the Deletion for all metrics except CAM that we did not have CAM for.

| | Gradient-based | | | CAM-based | | Perturbation-based | | | |
|---|---|---|---|---|---|---|---|---|---|
| | **Gradient** | **IG** | **SHAP** | **GradCAM** | **CAM** | **RISE** | **LIME** | **MP** | **EP** |
| **AlexNet** | 0.027 | 0.020 | 0.019 | 0.036 | N/A | 0.029 | 0.047 | 0.039 | 0.037 |
| **AlexNet-R** | **0.014** | **0.009** | **0.008** | **0.014** | N/A | **0.010** | **0.010** | **0.017** | **0.014** |
| **GoogleNet** | 0.045 | 0.033 | 0.031 | 0.036 | 0.036 | 0.033 | 0.043 | 0.041 | 0.038 |
| **GoogleNet-R** | 0.031 | 0.017 | 0.015 | 0.024 | **0.023** | 0.020 | 0.030 | 0.031 | 0.024 |
| **ResNet** | 0.075 | 0.052 | 0.048 | 0.070 | 0.070 | 0.063 | 0.066 | 0.073 | 0.073 |
| **ResNet-R** | 0.044 | 0.026 | 0.022 | 0.031 | 0.032 | 0.026 | 0.040 | 0.043 | 0.033 |
| **MobileNet** | 0.052 | 0.039 | 0.038 | 0.063 | N/A | 0.054 | 0.060 | 0.068 | 0.066 |
| **MobileNet-R** | 0.023 | 0.017 | 0.013 | 0.018 | N/A | 0.015 | 0.015 | 0.021 | 0.019 |
| **DenseNet** | 0.114 | 0.086 | 0.082 | 0.078 | 0.078 | 0.089 | 0.077 | 0.091 | 0.085 |
| **DenseNet-R** | 0.042 | 0.031 | 0.024 | 0.035 | 0.035 | 0.031 | 0.033 | 0.044 | 0.038 |
| **PGD-5** | 0.081 | 0.057 | 0.063 | 0.071 | 0.072 | 0.068 | 0.075 | 0.086 | 0.070 |
| **PGD-1** | 0.070 | 0.049 | 0.054 | 0.068 | 0.072 | 0.065 | 0.074 | 0.082 | 0.072 |

Table 16: Correlation between four metrics on ImageNet. For ImageNet-CL correlation see Table 8b.

| | **Pointing game** | **WSL** | **Insertion** | **Deletion** |
|---|---|---|---|---|
| **Pointing game** | 1 | 0.81 | 0.38 | 0.15 |
| **WSL** | | 1 | 0.30 | 0.15 |
| **Insertion** | | | 1 | 0.90 |
| **Deletion** | | | | 1 |

### A.7 Heatmaps

This section is an extension for Fig. 3 to demonstrate the heatmaps of AM methods of different sample images from ImageNet-CL.

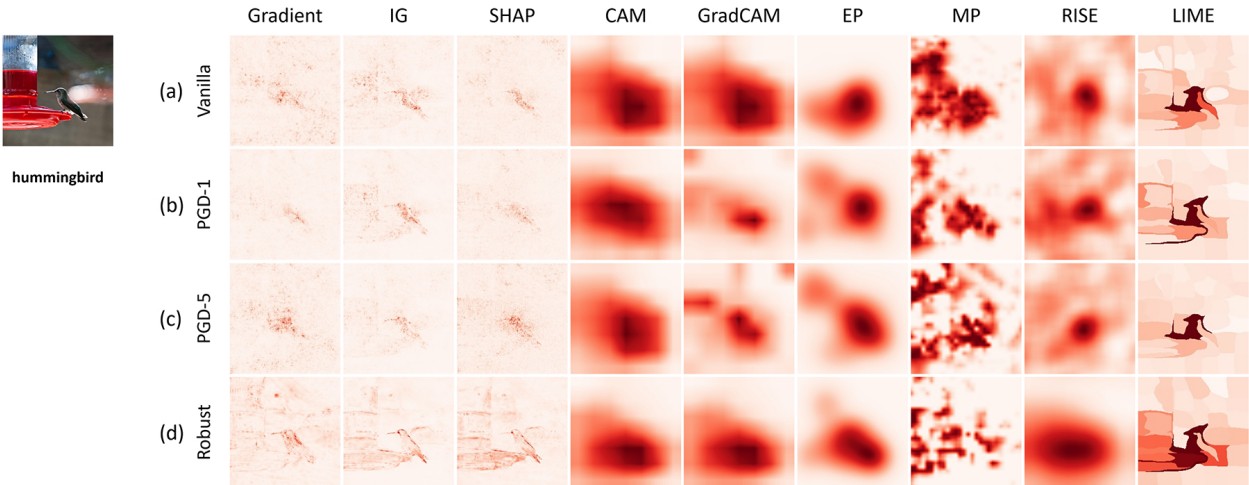

Figure 17: Heatmaps of 9 AM methods for four different CNNs of ResNet-50 architecture for the same input image of class "hummingbird". From top down: (a) vanilla ImageNet-trained ResNet-50 (He et al., 2016); (b–c) the same architecture but trained using AdvProp (Xie et al., 2020) where adversarial images are generated using PGD-1 and PGD-5 (i.e. 1 or 5 PGD attack steps (Madry et al., 2017) for generating each adversarial image); and (d) a robust model trained exclusively on adversarial data via the PGD framework (Madry et al., 2017). We can see the same trend as we saw in Fig. 3 from top down, gradient-based methods' AM are less noisy and more interpretable.

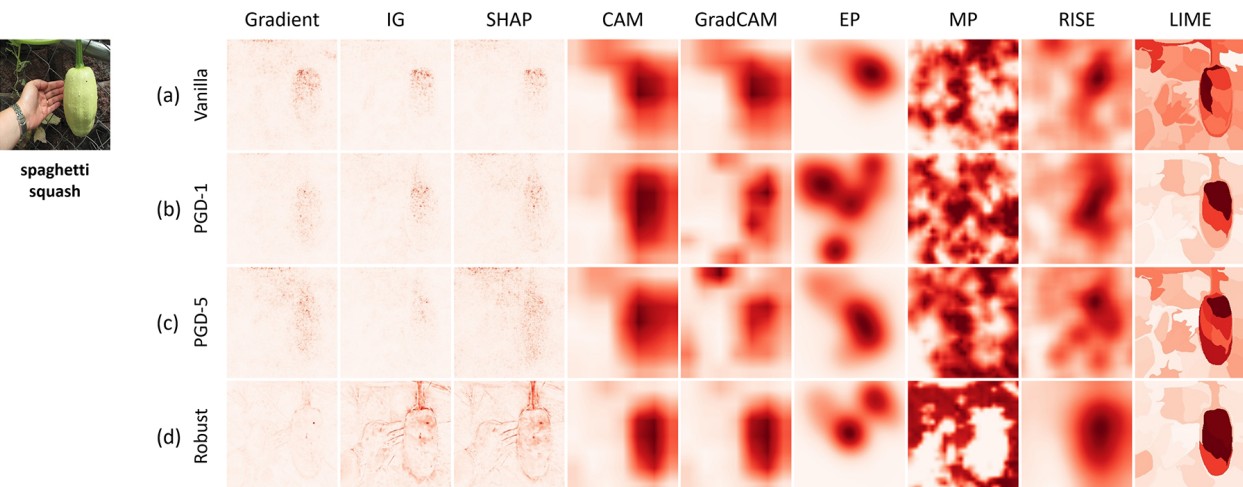

Figure 18: Heatmaps of 9 AM methods for four different CNNs of ResNet-50 architecture for the same input image of class "spaghetti squash". From top down: (a) vanilla ImageNet-trained ResNet-50 (He et al., 2016); (b–c) the same architecture but trained using AdvProp (Xie et al., 2020) where adversarial images are generated using PGD-1 and PGD-5 (i.e. 1 or 5 PGD attack steps (Madry et al., 2017) for generating each adversarial image); and (d) a robust model trained exclusively on adversarial data via the PGD framework (Madry et al., 2017). We can see the same trend as we saw in Fig. 3 from top down, gradient-based methods' AM are less noisy and more interpretable.

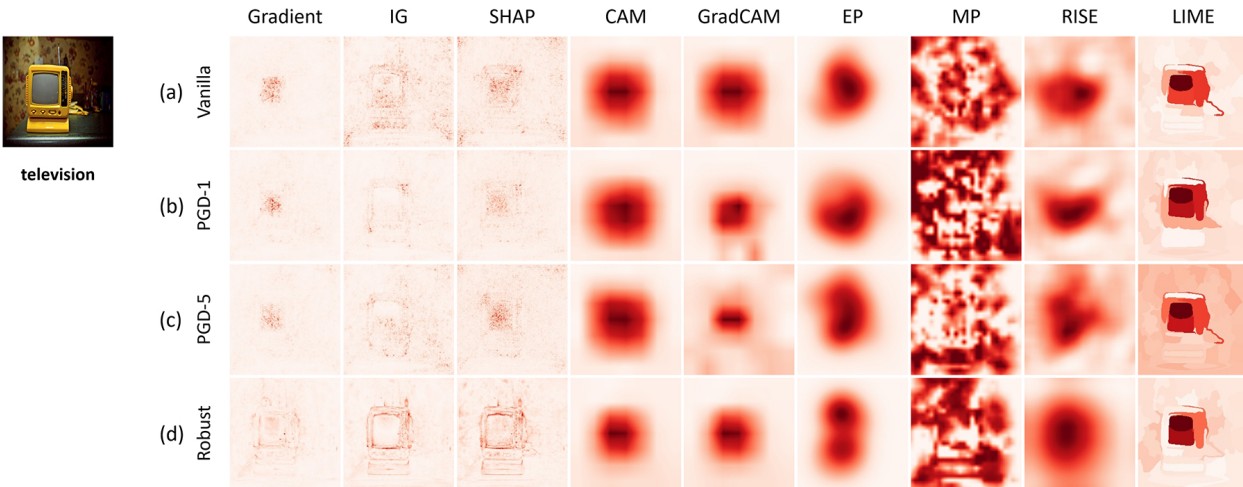

Figure 19: Heatmaps of 9 AM methods for four different CNNs of ResNet-50 architecture for the same input image of class "television". From top down: (a) vanilla ImageNet-trained ResNet-50 (He et al., 2016); (b–c) the same architecture but trained using AdvProp (Xie et al., 2020) where adversarial images are generated using PGD-1 and PGD-5 (i.e. 1 or 5 PGD attack steps (Madry et al., 2017) for generating each adversarial image); and (d) a robust model trained exclusively on adversarial data via the PGD framework (Madry et al., 2017). We can see the same trend as we saw in Fig. 3 from top down, gradient-based methods' AM are less noisy and more interpretable.

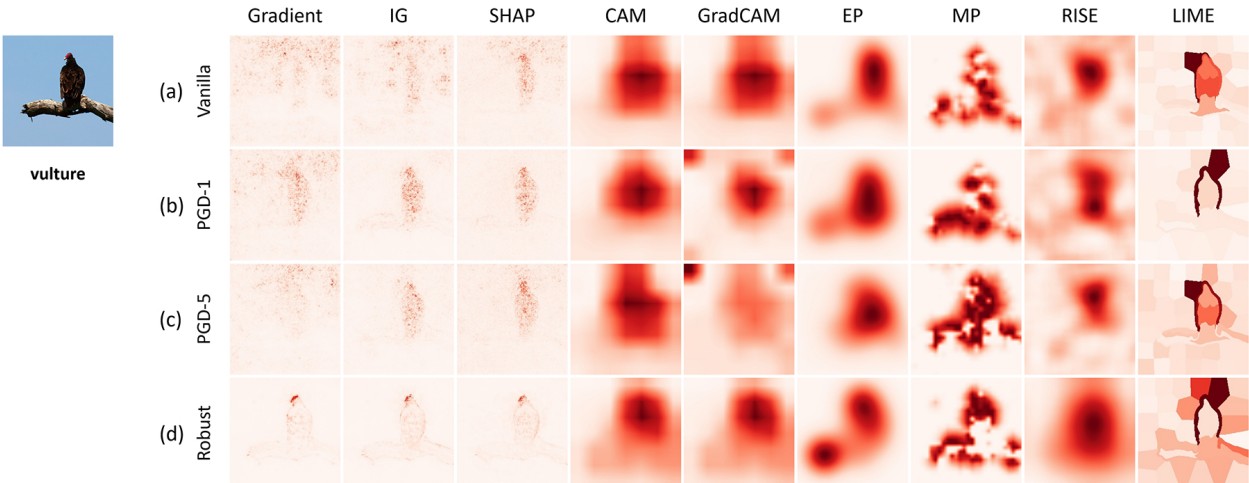

Figure 20: Heatmaps of 9 AM methods for four different CNNs of ResNet-50 architecture for the same input image of class "vulture". From top down: (a) vanilla ImageNet-trained ResNet-50 (He et al., 2016); (b–c) the same architecture but trained using AdvProp (Xie et al., 2020) where adversarial images are generated using PGD-1 and PGD-5 (i.e. 1 or 5 PGD attack steps (Madry et al., 2017) for generating each adversarial image); and (d) a robust model trained exclusively on adversarial data via the PGD framework (Madry et al., 2017). We can see the same trend as we saw in Fig. 3 from top down, gradient-based methods' AM are less noisy and more interpretable.

