# OpenReview forum: "How explainable are adversarially-robust CNNs?"
_TMLR — Rejected by TMLR_

### Review · Reviewer_uqq6 · 2023-04-11

**Summary Of Contributions:**

The paper studies three criteria to evaluate neural networks: test-set accuracy, out-of-distribution accuracy and explainability.  Specifically, it conducts experiments by jointly combining two criteria on different model architectures and training algorithms. It selects a wide range of interpretation methods from gradient-based, CAM-based, perturbation-based and AM methods and different training methods from adversarial training, advProp and normal training. Empirically, it first shows robust models provides a better score over gradient-based and AM methods so that CAM-based method is recommended to use. Second, it shows the advProp could achieve better test-set performance and explainability. It then studies which interpretation method could yield a best score on several score functions. It also studies the scores difference on different localization metrics, correlation with the confidence score etc.

**Audience:**

Yes

**Broader Impact Concerns:**

I don't have any ethical concerns.

**Claims And Evidence:**

No

**Requested Changes:**

1. Check the model performance of AdvProp and adversarial training.
2. Rearrange the experiments and findings in a more organized way.
3. Add some claims or assumption about the golden standard used in every experiments.

**Strengths And Weaknesses:**

Pros:
1. The paper conduct comprehensive experiments to study the intersection between three widely used criteria. It shows some interesting findings and also gives some useful suggestions.


Cons:
1. The paper's organization needs to be improved. The paper covers a lot of studies with different purposes. However, the order of different findings looks very random to me.
2. The paper didn't have a golden standard or ground truth among their studies. Some suggestions are given by the best performance got by some metrics. However, it is not clear whether those metrics could be truly represent the explainability.
3. The robust accuracy on AdvProp in Table 1 looks wrong to me. AdvProp couldn't improve the model's robustness as far as I know and it is nearly impossible that it will outperform adversarial training in terms of robust accuracy.

---

> ### Author Response · Authors · 2023-06-03
> **Response to Reviewer uqq6**
>
> ## > "Check the model performance of AdvProp and adversarial training."
>
> We note that the original CVPR 2020 paper [1] reported in Table 4 that AdvProp did improve both the test-set _and_ adversarial accuracy for large models (EfficientNet-B7) but not standard models such as ResNet-50 (i.e. they reported that, when training using AdvProp, test-set accuracy is low in exchange for high adversarial accuracy).
> However, we later empirically found that AdvProp with proper hyperparameter tuning CAN _also improve both_ test-set and adversarial accuracy. This is consistent with the result reported in an [independent project on Github](https://github.com/tingxueronghua/pytorch-classification-advprop).
>
> In our paper, we report the OOD accuracy on the validation set of ImageNet with the PGD attacks of seven steps per image; $L_2$ norm $\epsilon$ = 3, and a step size = 0.5.
> For validation and reproduction, our anonymous test code can be found [here](https://anonymous.4open.science/r/AdvProp-sample).
>
>
> ## > "Rearrange the experiments and findings in a more organized way."
>
> Thank you for this feedback!
> In light of your comment, we add a new Figure 1 to illustrate the story of our paper and provide a list of concrete questions (at the beginning of Sec. 3).
>
> - We first answer the paper's central question of how adversarial training affect a network's feature-attribution maps (Sec. 3.1).
> - Then, we address the question of whether feature-attribution methods are similar when averaged over network architectures and training regimes (Sec. 3.2). Interestingly, GradCAM and RISE consistently are the better ones.
> - We then compare network architectures based on their attribution map quality and found ResNet-50 to be the all-around winner (Sec. 3.3).
> - Next, we compare models based on three important properties (explainability via feature attribution, number of multiply-accumulate operations, and classification accuracy on real images) and the possible trade-offs practitioners should make (Sec. 3.4).
> - From the same study over 5 network architectures, 3 training algorithms, and 9 feature-attribution methods, we find that AdvProp results in the models that are on average, the better training paradigm compared to training on real or adversarial data alone (Sec. 3.5). That is, AdvProp models tend to outperform vanilla models on both classification accuracy and feature-map explainability.
> - Finally, interestingly, our novel comparison between vanilla and adversarially-robust models reveal important findings that the common Insertion and Deletion metrics strongly correlate with confidence scores (Sec. 3.6) and weakly correlate with the other common metric of weakly-supervised localization (Sec. 3.7).
>
> ------------------------------
> [1] Xie, C., Tan, M., Gong, B., Wang, J., Yuille, A.L. and Le, Q.V., 2020. Adversarial examples improve image recognition. In Proceedings of the IEEE/CVF Conference on Computer Vision and Pattern Recognition (pp. 819-828).

---

> ### Author Response · Authors · 2023-06-03
> **Response to Reviewer uqq6**
>
>
> ## > "Add some claims or assumption about the golden standard used in every experiments."
>
> **Definition of Explainability**: We wish to clarify that we focus on explanability of CNNs via _feature-attribution only_ because feature attribution is among the most mature directions in Explainable AI with hundreds of papers in the last few years.
>
> In evaluating feature attribution maps, we use a diverse set of four different, common metrics (Sec. 2.4) and provide the assumptions behind these standard metrics.
>
> **Our new text in Sec. 2.4 reads as the following:**
>
> > _Assumptions of PG and WSL:_ Both PG and WSL are standard metrics in evaluating AMs [2,3,4]. While PG is a more coarse metric and WSL a more fine-grained version, both are based on the assumption that the discriminative signals for CNNs to label an image are on the main object annotated by humans.
>
> > _Assumptions of Deletion and Insertion:_ The Deletion and Insertion metrics [5] are based on two assumptions. First, it assumes that each pixel is an independent variable. Therefore, it is possible to change a model's predictions by removing or adding one pixel. Second, it assumes that removing a pixel by replacing it with a zero (i.e. gray) pixel is a reasonable removal operator that yields a counterfactual sample near or on the true data manifold [6].
>
> **In Sec. 5 (Limitations), we also add the acknowledgment that automatic evaluation metrics have their own limitations and are not equivalent to a more direct test on humans:**
>
> > We acknowledge that the most _objective_ evaluation metrics for attribution maps should be measuring how attribution maps improve human performance on a downstream task (e.g. efficiency or accuracy) [7]. Yet, performing a human study for each triplet of (training algorithm, network architecture, and attribution method) is extremely expensive and, therefore, out of the scope of this paper.
> Instead, we hope to shed light on model selection based on the automatic-evaluation metrics (PG, WSL, Insertion, and Deletion).
>
>
> ------------------------------
>
> [2] Bolei Zhou, Aditya Khosla, Agata Lapedriza, Aude Oliva, and Antonio Torralba. Learning deep features
> for discriminative localization. In Proceedings of the IEEE conference on computer vision and pattern
> recognition, pp. 2921–2929, 2016.
>
> [3] Julius Adebayo, Justin Gilmer, Michael Muelly, Ian Goodfellow, Moritz Hardt, and Been Kim. Sanity checks
> for saliency maps. arXiv preprint arXiv:1810.03292, 2018.
>
> [4] David Bau, Bolei Zhou, Aditya Khosla, Aude Oliva, and Antonio Torralba. Network dissection: Quantifying
> interpretability of deep visual representations. In Proceedings of the IEEE conference on computer vision
> and pattern recognition, pp. 6541–6549, 2017.
>
> [5] Vitali Petsiuk, Abir Das, and Kate Saenko. Rise: Randomized input sampling for explanation of black-box
> models. arXiv preprint arXiv:1806.07421, 2018
>
> [6] Ian Covert, Scott Lundberg, and Su-In Lee. Feature removal is a unifying principle for model explanation methods.
>
> [7] Giang Nguyen, Anh Nguyen. The effectiveness of feature attribution methods and its correlation with automatic evaluation scores. NeurIPS 2021.

---

### Review · Reviewer_UZQW · 2023-04-13

**Summary Of Contributions:**

The paper consists in a detailed experimental comparison of explainability mechanisms for deep networks over images. It compares several gradient-based (IG, SHAP), CAM-based (CAM, GradCAM) and perturbation-based (MP, EP, LIME, RISE) explainability techniques with different metrics (PG, WSL, Insertion, Deletion) for comparing with robust (adversarially trained) models, feature attribution, accuracy, architecture choices, and relevance of Insertion and Deletion scores. The experiments are conducted in a rather systematic way, with results covering various aspects of interest for assessing the performance of these explainability techniques.

**Audience:**

Yes

**Broader Impact Concerns:**

I don’t see any specific ethical impacts in that work. Assessing explainability in deep models could in fact support transparency in AI, which is of relevance in varied ethical contexts.

**Claims And Evidence:**

Yes

**Requested Changes:**

I would invite the authors to provide a more detailed overview and synthesis of their results in the limitation and conclusion section. I found the current sections 5 and 6 to be rather short and not carrying much, I would have liked to get something more elaborated. The six major findings at the end of section 1 are relevant and very useful, but once someone has gone through all the paper, it may be interesting to position this more generally and provide a clear and insightful take-home message in sections 5 and 6.

**Strengths And Weaknesses:**

Strengths:
- The experiments appear to be well executed and cover different aspects of interest for assessing explainability mechanisms of neural models processing images.
- Explainability is an important topic in machine learning, as we need to provide ways to understand how decisions are made and ensure that the models are working properly.
- The overall organization of the paper is good, with professional-looking figures and tables and careful layout.

Weaknesses:
- The paper is purely experimental with existing methods, it contains no original technical contributions, nor theoretical ones.
- The results are overall not very surprising, except maybe for the deletion score that appears not to be a recommended metric for explainability. Some attribution mechanisms as EP, GradCAM and RISE appear to outperform the other approaches, but the difference with other approaches remains relatively narrow.
- The writing style is not always very clear, it took me several readings to gain a good understanding of the content presented in the paper, the paper tends to be rather enumerative in its approach, with explanations that can be rather specific. I have difficulties extracting the big picture of the paper.

---

> ### Author Response · Authors · 2023-06-03
> **Response to Reviewer UZQW**
>
> ## > "Provide a more detailed overview and synthesis of their results in the limitation and conclusion section."
>
> Thank you for the spot-on suggestion! Per your request, we have made major revisions to the Limitations (Sec. 5) and Conclusion (Sec. 6) sections to include a thorough list of insights and perspectives to better inform readers of the novelty, pros and cons of our study.
>
> In light of your feedback, our Sec. 5 (Limitations) has been revised to contain a thorough discussion of the feature-attribution evaluation metrics:
>
> > **Feature-attribution evaluation**
> The surprising results in Secs.3.6 and 3.7 show that how to _automatically_ evaluate feature attribution maps is still an open problem and despite being popular, PG, WSL, Insert, and Deletion may measure an undesired property of the network.
>
> > We acknowledge that one of the most _objective_ evaluation metrics for attribution maps should be measuring how attribution maps improve human performance (e.g. efficiency or accuracy) on a downstream task [1]. Yet, performing a human study for each triplet of (training algorithm, network architecture, and attribution method) in our large-scale study is prohibitively expensive and, therefore, out of the scope of this paper.
> Instead, we hope to shed light on the model selection based on the automatic evaluation metrics (PG, WSL, Insertion, and Deletion).
>
> We also refactored Sec. 6 to contain a story of the major findings. Our Conclusion section (Sec.6) now reads as the following:
> > We show that robust models under gradient-based methods are significantly and consistently more explainable compared to their vanilla counterparts; however, such patterns were not seen in other categories of AMs, i.e., perturbation-based and CAM-based. Analyzing AdvProp models showed that even though they achieve higher in and out of distribution accuracies, they could not outperform adversarially robust trained models in terms of explainability. Evaluating attribution methods across all 12 CNNs revealed that GradCAM, although introduced in 2017, still seems to be the best overall AM method taking into account both runtime and overall performance of the methods across 12 networks and four metrics which somehow coincides with the results in \citep{choe2020evaluating}. Also, it should be noted that CAM-based methods suggest almost no difference in explainability ability between robust and vanilla models which suggests that choosing the CAM-based methods is a safe choice in both scenarios.
>
> > Furthermore, interestingly, PGD-1 models perform roughly the best under PG, WSL, and Insertion on ImageNet images (Fig. 16). The overall promising results of Advprop invite future research on how to leverage AdvProp models for the actual downstream human-in-the-loop image classification tasks.
>
> > Overall, this study shed light on the strengths, similarities, and limitations of different vanilla and robust CNNs across various architectures and attribution methods, emphasizing the need to strike a balance between model accuracy and explainability. The findings provide valuable insights and recommendations for researchers and practitioners working on explainable AI on how to use attribution methods better and efficiently, which encourages further exploration and advancements in this field.
> ------------------------------
>
> [1] Giang Nguyen, Anh Nguyen. The effectiveness of feature attribution methods and its correlation with automatic evaluation scores. NeurIPS 2021.

---

### Review · Reviewer_PBY1 · 2023-05-19

**Summary Of Contributions:**

This work is the first large-scale evaluation of many CNNs and many methods, in which it covers three main sets of representative methods: gradient-based, perturbation-based, and CAM-based. The results showed that robust models under gradient-based methods are significantly more explainable compared to their vanilla counterparts; however, such patterns were not seen in other categories of AMs.

**Audience:**

Yes

**Claims And Evidence:**

Yes

**Requested Changes:**

Address my concerns detailed in Weaknesses.

**Strengths And Weaknesses:**

Strengths

- This paper has conducted a thorough empirical study.
- The gained conclusions are convincing and insightful.

Weaknesses
- The scope of the experiments is still limited. For example, can you experiment on more recent architectures such as ViT and more general cases (such as language modeling)?

- The mentioned ''out-of-distribution'' finally boils down to accuracy under adversarial attacks? The consideration of natural robustness seems to be more reasonable.

- Why are the attribution methods mainly considered? Can you incorporate other explanation tools to draw a more rigorous conclusion?

---

> ### Author Response · Authors · 2023-06-03
> **Response to Reviewer PBY1**
>
> ## > "The scope of the experiments is still limited. For example, can you experiment on more recent architectures such as ViT and more general cases (such as language modeling)?"
>
> Thank you for your suggestion! Our study is solely focused on CNNs, which is the default, mature testbed for feature-attribution methods, many of which are designed for CNNs exclusively (e.g., CAM and GradCAM). Adapting feature attribution to ViTs require modifications to the algorithms [1] and sometimes also the models [2] and therefore are out of the scope of our paper.
>
> To the best of our knowledge, our study is the largest-scale study in Explainable AI in terms of the inclusion of methods, network architectures, and training algorithms. Specifically, we test **9** feature-attribution methods, **5** network architectures, **12** different classifiers, and **3** training algorithms. We capture this unprecedented breadth of our study in a newly created Fig. 1.
>
> We believe testing on Transformers and Language models is an interesting direction for follow-up work.
>
> -------------
> [1] Transformer Interpretability Beyond Attention Visualization. CVPR 2021
>
> [2] Generic Attention-model Explainability for Interpreting Bi-Modal and Encoder-Decoder Transformers. ICCV 2021
>
>
> ## > "The mentioned ''out-of-distribution'' finally boils down to accuracy under adversarial attacks? The consideration of natural robustness seems to be more reasonable."
>
> Yes, the out-of-distribution accuracy in this paper refers to accuracy under pixel-wise adversarial attacks. The motivation for this focus is two-fold:
> 1. Prior work has shown that explicitly smoothing attribution maps by averaging over gradients at pixel-wise noisy images creates more clean and interpretable heatmaps [1] [2]. Furthermore, Bansal et al. [2] showed that adversarially-trained CNNs also tend to admit clean, interpretable heatmaps.
>
> 2. Adversarial training (Madry et al. 2017) and AdvProp are two state-of-the-art approaches in training models to be resistant to out-of-distribution pixel-wise noisy data (adversarial examples). Therefore, we include these two training regimes in our study (in addition to the standard training on real images via the cross-entropy classification loss). We note that AdvProp models tend to outperform vanilla models on also natural adversarial images, e.g. ImageNet-C [3][4].
>
>
> Due to the above reasons, we chose adversarial accuracy to be the representative out-of-distribution accuracy in this paper. "Natural robustness" is an interesting topic but also fairly broad. There are many benchmarks (e.g. ImageNet-A, ImageNet-C, ImageNet-Sketch, Strike-With-A-Pose), each measure robustness in a different perspective. If you think there is a benchmark that fits the bill better, please let us know. We'd be happy to consider including it in our paper.
>
> -------------------------
> [1] SmoothGrad: removing noise by adding noise. Smilkov et al. 2017
>
> [2] SAM: The Sensitivity of Attribution Methods to Hyperparameters. CVPR 2020
>
> [3] Adversarial Examples Improve Image Recognition. CVPR 2020
>
> [4] The shape and simplicity biases of adversarially robust ImageNet-trained CNNs. WHI 2020
>
>
> ## > "Why are the attribution methods mainly considered? Can you incorporate other explanation tools to draw a more rigorous conclusion?"
>
> Thank you for a great question!
> In this work, we focus on explanations via feature attribution methods, which is among the most mature tool in Explainable AI, with hundreds of publications since 2014. Different explanation modalities (example-based, tree-based, patch-based, etc..) have different evaluation metrics and therefore are not directly comparable. Also, it is largely unknown how adversarial training affects non-heatmap-based explanations.
> Thus, we are leaving other explanation types for future work.

---

### Author Response · Authors · 2023-06-03
**Response to all reviewers**

Thank you all very much for your time and constructive comments that have strengthened our paper! We have heavily edited the manuscript in light of your feedback.
- We added a new figure (Figure 1) to capture the motivation for our research, and our series of experiments.
- We make major revised textual changes blue to aid your review (the final text color will be black following TMLR style).
- We released anonymous code to aid the reproduction of the AdvProp training (in reply to a question by reviewer `uqq6`)

**Organization and motivation of our study**

In light of your feedback, we add a new Figure 1 to illustrate the story of our paper and provide a list of concrete questions (at the beginning of Sec. 3).

- We first answer the paper's central question of how adversarial training affects a network's feature-attribution maps (Sec. 3.1).
- Then, we address the question of whether feature-attribution methods are similar when averaged over network architectures and training regimes (Sec. 3.2). Interestingly, GradCAM and RISE consistently are the better ones.
- We then compare network architectures based on their attribution map quality and found ResNet-50 to be the all-around winner (Sec. 3.3).
- Next, we compare models based on three important properties (explainability via feature attribution, number of multiply-accumulate operations, and classification accuracy on real images) and the possible trade-offs practitioners should make (Sec. 3.4).
- From the same study over 5 network architectures, 3 training algorithms, and 9 feature-attribution methods, we find that AdvProp results in the models that are, on average, better than those trained exclusively on real images or adversarial images alone (Sec. 3.5). That is, AdvProp models tend to outperform vanilla models on both classification accuracy and feature-map explainability.
- Finally, interestingly, our novel comparison between vanilla and adversarially-robust models reveals important findings that the common Insertion and Deletion metrics strongly _correlate_ with confidence scores (Sec. 3.6) and _weakly_ correlate with the other common metric of weakly-supervised localization (Sec. 3.7).

Below are our inline responses to each reviewer's comments and requested changes.

---

### Decision · Action_Editors · 2023-06-21

**Recommendation:** Reject

**Comment:**

For my part, I have to issue a concurring-in-judgement style opinion.

Reviewing the discussion and reading the concerns of the reviewers, I don't feel as though the critiques the reviewers have remaining are adequate critiques, in the sense that I don't feel as though they are material for deciding whether this paper adequately supports its claims.  The remaining reviewer concerns seem to be (1) the paper is a bit disorganized (2) the paper lacks theoretical analysis and (3) the paper focuses on CNNs rather than transformers.

I do think the paper was made more clear by the authors after the edit, taking a bit step to addressing (1), and as for (2) and (3) I don't think those really should matter for accessing whether this paper supports its own claims within its own scope.

Unfortunately, while I disagree with the reasons the reviewers give for recommending rejection, I find myself in a concurring position of agreeing with the final judgement.

The paper constitutes a large empirical study into various issues surrounding explainability of CNNs, and I commend the authors for the work and experiments they have done.  At the same time I find the analysis very disappointing, to the point that I don't think the evidence provided in the paper supports the claims being made.

Basically, I think the paper is over-claiming due to broken analysis. To give just a couple examples of what I mean, consider the Figure 2 and the first sentence of the caption.  "The AMs of robust models consistently score higher than those of vanilla models on Pointing Game."

The evidence to support this claim is provided by Figure 2 (a), where we see scores on the pointing game for both the robust and vanilla variants along with their standard deviations.  Looking at Figure 2 (a), its immediately clear that those intervals basically all overlap and we should be very hesitant of any sort of definitive claim, and yet the paper seems to interpret the results as "consistently scoring higher".  It also has a p-value on the plot of 0.000009. I'm generally suspicious of small p values to begin with and in this case with the scores and intervals present, I'm extremely skeptical that that p-value really means what it purports to mean.  The first and last bar plots are clearly overlapping.  The middle bar plot shows the two intervals just outside the 1 sigma boundary (I'm assuming the bars are showing the standard deviations though its not stated for this plot, as it is stated elsewhere in the paper).  For bars like this, if we sampled values from two Gaussian distributions with the given means and standard deviations, we shouldn't expect that the blue bar value was larger than the red barred value more than 9 times out of 10 or so.  I was able to verify this by digitizing the plot and sampling from normals with the observed means and standard deviations myself.  I wouldn't say that this gives strong enough evidence to state that the models 'consistently score higher' than the alternative.   A statement like that made with the evidence provided in Figure 2 a really tests my credibility, it feels very unreasonable.

A lot of the results in the paper are like this.  For instance, look at Figure 4(c), look at those bars and associated standard deviations.  Imagine I removed all of the labels and the context was lacking.  All you know is that I've measured 7 different populations and observed 7 different means and standard deviations.  Would you be able to look at that plot and believe me if I said that there was a clear winner and loser amongst the set of 7 populations?

Looking at Table 4 in the appendix, it seems clear what went wrong.  A lot of experiments were run, a lot of results were collected and the performance of each method was then summarized with a single set of summary statistics, the mean and standard deviation, but then all judgments made on the different methods then seemed to only consider the mean performance without regard to the spread.  This does a great disservice to the work the paper has done in collecting all of those experiments.  To reduce all of the experiments to a single rank amongst 7 means really ignores all of the signal present in the data.  Its clear from the "average of rankings" column in Table 4 that claims in the paper like "GradCAM and RISE are the best feature attribution methods" are based on the observation that the mean rank for those methods are 1st, 3rd, 2nd, and 3rd for an average ranking of 2.25.  Does this really feel like an honest evaluation of the performance of the methods?

Looking at the spread of values in the first column for instance, RISE is given the rank of first, with a mean of 91.41 and a standard deviation of 1.87, but GradCAM has a mean of 85.32 with a standard deviation of 10.19.  If we trust the normal approximation, sampling from those two distributions, the sample from the RISE distribution would only be larger than the GradCAM 5 out of every 7 times.  If I repeat this for the other entries in the column and draw samples from the corresponding normal approximations, its true that RISE is ranked first more than other methods (because it has a higher mean), but it comes out on top less than half of the time (49%).  To be ranked first less than half the time and for all of that variability to be ignored and for RISE to then be treated as the best for the pointing game really does discredit to all of the evidence collected.  If I repeat this for all four of the tables and compute a version of the average rankings that incorporates this notion of measured variability, the top line result actually changes.  It is actually RISE and EP, not RISE and GradCAM that perform the best.   And this is simply looking at something like average ranks, rather than trying to treat each of the various games as something like an opinion about which methods are best and applying something like a voting method to aggregate the scores.  If you do that, using 200k samples from the approximate normals summarized in the tables, you find that RISE is a Condorcet winner in this little simulated election, pairwise beating the other methods with EP again the second place finisher.  This isn't to say that I necessarily trust what I just did as a definitive answer for which methods are best either.  Reading the short description of what I did, you're likely skeptical about the claim I just made and that I think is my real point.  These results are not so definitive as to support a bold claim about which method is best, and certainly not to support the claim that "GradCAM and RISE are the best feature attribution methods".

I understand that the authors generated their p-values with a Mann-Whitney U Test, but without further details about the measure of central tendencies used, the value the statistic took, sample sizes used, etc, I'm not sure what has gone wrong there, but I don't believe the p-values stated in the paper.  More broadly, I don't think a p-value being a small value is the kind of evidence we as a community should take as "clear and convincing" evidence in and of itself.  If one method is better than another, it should be easy to show that and convince an otherwise skeptical reader without appealing to named statistics tests.

Honestly, the comparisons should be made at the level of the individual samples, rather than first summarizing the performance of each method by means of its mean and standard deviation.

Overall, I like the goal of the paper.  I think its noble to want to investigate various AM methods and try to assess and compare their performance, and it seems like a lot of data was collected in that quest, but I think essentially all of the analysis done on that data needs to be redone, as right now the paper does a great disservice to its own data in using such imprecise analysis and in so doing commits the sin of making claims that its own figures and data cannot support.

For these reasons I have to vote to reject the paper at this time, but honoring the work that has been done, I am going to allow the authors to consider resubmitting a major revision at a later time, but I want to warn the authors that I expect essentially a complete redoing of all of the analysis, the statistics need to be started afresh.

**Audience:**

The paper clearly has an audience at TMLR.

**Claims And Evidence:**

Initially, the reviewers had concerns about whether the paper raised to the requirement of TMLR that its claims are backed up by clear, convincing, and accurate evidence.

The authors, to their credit, responded to the criticism and made changes to the paper to improve its clarity.

Unfortunately, the reviewers didn't then further engage with the authors to express any remaining or lingering concerns they had.  Though, after reviewing the authors responses, each of the reviewers made a recommendation as to whether the paper should be accepted, and presently two of the three reviewers are recommending rejection and have indicated that they don't feel as though the paper meets the claims and evidence threshold for TMLR.

**Resubmission Of Major Revision:**

The authors may consider submitting a major revision at a later time.